# Coevolution of the CDCA7-HELLS ICF-related nucleosome remodeling complex and DNA methyltransferases

**Hironori Funabiki[1]\*, Isabel E Wassing[1], Qingyuan Jia[1], Ji-Dung Luo[2], Thomas Carroll[2]**

[1]Laboratory of Chromosome and Cell Biology, The Rockefeller University, New York, United States; [2]Bioinformatics Resource Center, The Rockefeller University, New York, United States

**\*For correspondence:** funabih@rockefeller.edu

**Competing interest:** The authors declare that no competing interests exist.

**Sent for Review** 08 March 2023
**Preprint posted** 09 March 2023
**Reviewed preprint posted** 05 June 2023
**Reviewed preprint revised** 11 September 2023
**Reviewed preprint revised** 19 September 2023
**Version of Record published** 28 September 2023

**Abstract** 5-Methylcytosine (5mC) and DNA methyltransferases (DNMTs) are broadly conserved in eukaryotes but are also frequently lost during evolution. The mammalian SNF2 family ATPase HELLS and its plant ortholog DDM1 are critical for maintaining 5mC. Mutations in HELLS, its activator CDCA7, and the de novo DNA methyltransferase DNMT3B, cause immunodeficiency-centromeric instability-facial anomalies (ICF) syndrome, a genetic disorder associated with the loss of DNA methylation. We here examine the coevolution of CDCA7, HELLS and DNMTs. While DNMT3, the maintenance DNA methyltransferase DNMT1, HELLS, and CDCA7 are all highly conserved in vertebrates and green plants, they are frequently co-lost in other evolutionary clades. The presence-absence patterns of these genes are not random; almost all CDCA7 harboring eukaryote species also have HELLS and DNMT1 (or another maintenance methyltransferase, DNMT5). Coevolution of presence-absence patterns (CoPAP) analysis in Ecdysozoa further indicates coevolutionary linkages among CDCA7, HELLS, DNMT1 and its activator UHRF1. We hypothesize that CDCA7 becomes dispensable in species that lost HELLS or DNA methylation, and/or the loss of CDCA7 triggers the replacement of DNA methylation by other chromatin regulation mechanisms. Our study suggests that a unique specialized role of CDCA7 in HELLS-dependent DNA methylation maintenance is broadly inherited from the last eukaryotic common ancestor.

## eLife assessment

This **important** manuscript reveals signatures of co-evolution of two nucleosome remodeling factors, Lsh/HELLS and CDCA7, which are involved in the regulation of eukaryotic DNA methylation. The results suggest that the roles for the two factors in DNA methylation maintenance pathways can be traced back to the last eukaryotic common ancestor and that the CDC7A-HELLS-DNMT axis shaped the evolutionary retention of DNA methylation in eukaryotes. The **solid** evolutionary analyses form a strong basis for experimental follow-up studies. The work should be of interest to colleagues in the fields of evolutionary biology, chromatin biology and genome biology.

## Introduction

DNA methylation, particularly 5-methylcytosine (5mC) at CpG sequences, is widely conserved in eukaryotes. Along with its role in silencing transposable elements and suppressing aberrant intragenic transcription (*Choi et al., 2020*; *Deniz et al., 2019*; *Neri et al., 2017*), DNA methylation plays critical roles in developmental control, genome stability and the development of diseases such as cancers and immunodeficiencies (*Greenberg and Bourc'his, 2019*; *Lyko, 2018*; *Nishiyama and Nakanishi, 2021*;

*Robertson, 2005*). Despite its versatility as an epigenetic control mechanism, DNA methyltransferases (DNMTs) are lost in multiple evolutionary lineages (*Bewick et al., 2017b*; *Huff and Zilberman, 2014*; *Kyger et al., 2021*; *Ponger and Li, 2005*; *Zemach et al., 2010*). While the evolutionary preservation and loss of DNMTs and other proteins involved in 5mC metabolism has been studied (*Bewick et al., 2017b*; *de Mendoza et al., 2018*; *Dumesic et al., 2020*; *Engelhardt et al., 2022*; *Huff and Zilberman, 2014*; *Iyer et al., 2011*; *Lewis et al., 2020*; *Mondo et al., 2017*; *Mulholland et al., 2020*; *Nai et al., 2020*; *Tirot et al., 2021*; *Zemach et al., 2010*), it remains unclear if there is any common process or event that leads to the loss of DNA methylation systems in certain evolutionary lineages. Since eukaryotic genomes are compacted by nucleosomes, DNA methylation systems must deal with this structural impediment (*Felle et al., 2011*). Could the emergence or loss of a specific nucleosome regulator affect the evolution of DNA methylation as an epigenetic mechanism?

DNMTs are largely subdivided into maintenance DNMTs and de novo DNMTs (*Lyko, 2018*; *Ponger and Li, 2005*). Maintenance DNMTs (directly or indirectly) recognize hemimethylated CpGs and restore symmetric methylation at these sites to prevent the passive loss of 5mC upon DNA replication. Conversely, methylation by de novo DNMTs does not require methylated DNA templates. In animals, 5mC is maintained during DNA replication by DNMT1 together with UHRF1, which directly recognizes hemimethylated cytosine via the SRA domain and stimulates activity of DNMT1 in a manner dependent on its ubiquitin-ligase activity (*Nishiyama and Nakanishi, 2021*). De novo DNA methylation is primarily carried out by DNMT3 in animals, while plants encode the closely related de novo methyltransferase DRM (*Cao et al., 2000*). Some species, such as the fungus *Cryptococcus neoformans*, lack de novo DNA methylation activity (*Catania et al., 2020*; *Dumesic et al., 2020*; *Huff and Zilberman, 2014*). In *C. neoformans*, DNA methylation is maintained by DNMT5, which has a SNF2-family ATPase domain that is critical for its methyltransferase activity (*Dumesic et al., 2020*; *Huff and Zilberman, 2014*). Although DNMT5 orthologs cannot be found in land plants and animals, their broad existence in Stramenopiles, Chlorophyta and Fungi suggests that DNMT5, perhaps together with DNMT1, coexisted in the last eukaryotic common ancestor (LECA) (*Huff and Zilberman, 2014*).

DNA hypomethylation is a hallmark of immunodeficiency–centromeric instability–facial anomalies (ICF) syndrome, a rare genetic disorder which causes severe immune defects (*Ehrlich, 2003*; *Ehrlich et al., 2006*; *Vukic and Daxinger, 2019*). Activated lymphocytes of ICF patients display a characteristic cytogenetic abnormality at the juxtacentromeric heterochromatin of chromosome 1 and 16, where the satellite II repetitive element is highly enriched. In about 50% of ICF patients, classified as ICF1, the disease is caused by mutations in the de novo DNA methyltransferase DNMT3B (*Hansen et al., 1999*; *Okano et al., 1999*). The rarer genotypes known as ICF2, ICF3, and ICF4, are caused by mutations in ZBTB24, CDCA7, and HELLS (Helicase, Lymphoid Specific, also known as LSH, Lymphoid-Specific Helicase), respectively (*de Greef et al., 2011*; *Thijssen et al., 2015*; *Unoki, 2021*). While hypomethylation at satellite II repeats is common to all ICF genotypes, ICF2-4 patient-derived cells, but not ICF1 patient cells, additionally exhibit hypomethylation at centromeric alpha satellite repeats (*Jiang et al., 2005*; *Thijssen et al., 2015*; *Velasco et al., 2018*). Knock out of ICF genes in human HEK293 cells reproduces the DNA methylation profile observed in patient cells (*Unoki et al., 2019*). In mice, ZBTB24, HELLS, and CDCA7, but not DNMT3B, are required for methylation at centromeric minor satellite repeats (*Dennis et al., 2001*; *Hardikar et al., 2020*; *Ren et al., 2015*; *Thijssen et al., 2015*). Therefore, although all ICF proteins promote DNA methylation, the ZBTB24-CDCA7-HELLS axis may target additional loci such as alpha satellites for DNA methylation in a DNMT3-independent manner. Indeed, the importance of HELLS in DNA methylation maintenance by DNMT1 has been reported (*Han et al., 2020*; *Ming et al., 2020*; *Unoki, 2021*; *Unoki et al., 2020*).

HELLS belongs to one of ~25 subclasses of the SNF2-like ATPase family (*Flaus et al., 2006*). Among these diverse SNF2 family proteins, HELLS appears to have a specialized role in DNA methylation. Reduced genomic DNA methylation was observed in HELLS (LSH) knockout mice (*Dennis et al., 2001*), transformed mouse fibroblasts (*Dunican et al., 2013*), mouse embryonic fibroblasts (*Myant et al., 2011*; *Yu et al., 2014*), and zebrafish and *Xenopus* embryos (*Dunican et al., 2015*). In fact, this function of HELLS in DNA methylation was originally inferred from studies in *Arabidopsis*, where mutations in the HELLS ortholog DDM1 (Decrease in DNA Methylation) cause drastic reduction of 5mC in transposable and repetitive elements (*Miura et al., 2001*; *Vongs et al., 1993*). Like HELLS, DDM1 is a SNF2 ATPase with demonstrable in vitro nucleosome remodeling activity (*Brzeski and Jerzmanowski, 2003*; *Jenness et al., 2018*). Since DNA methylation defects in *ddm1* mutants

can be rescued by the loss of histone H1, it has been proposed that DDM1-mediated remodeling of H1-bound nucleosomes is important for DNA methylation (*Zemach et al., 2013*).

CDCA7 (also known as JPO1) was originally identified as one of eight CDCA genes that exhibited cell division cycle-associated gene expression profiles (*Walker, 2001*). A putative 4CXXC zinc finger binding domain (zf-4CXXC_R1) is conserved among CDCA7 homologs, including its paralog CDCA7L (also known as JPO2 and R1; *Chen et al., 2005*; *Ou et al., 2006*). Multiple lines of evidence support the idea that CDCA7 functions as a direct activator of the nucleosome remodeling enzyme HELLS. First, in *Xenopus* egg extracts, HELLS and CDCA7e (the sole CDCA7 paralog present in *Xenopus* eggs) both preferentially interact with nucleosomes rather than nucleosome-free DNA, and binding of HELLS to chromatin depends on CDCA7e (*Jenness et al., 2018*). Second, HELLS alone exhibits little nucleosome sliding activity, but CDCA7e greatly stimulates it (*Jenness et al., 2018*). Third, HELLS directly binds to CDCA7e (as well as CDCA7 and CDCA7L), even in the absence of DNA (*Jenness et al., 2018*). Fourth, HELLS and CDCA7 interact in human cells (*Unoki et al., 2019*), where chromatin binding of HELLS also depends on CDCA7 (*Jenness et al., 2018*). ICF disease mutations located in the conserved zf-4CXXC_R1 domain (R274C, R274H and R304H in human CDCA7) inhibited chromatin binding of CDCA7 and HELLS without interfering with CDCA7-HELLS interaction (*Jenness et al., 2018*), such that the zf-4CXXC_R1 domain likely serves as a chromatin binding module. Therefore, we proposed that CDCA7 and HELLS form a bipartite nucleosome remodeling complex, termed CHIRRC (<u>C</u>DCA7-<u>H</u>ELLS <u>I</u>CF-<u>r</u>elated nucleosome <u>r</u>emodeling <u>c</u>omplex) (*Jenness et al., 2018*).

We previously suggested that ZBTB24, CDCA7, and HELLS form a linear pathway to support DNA methylation (*Jenness et al., 2018*). ZBTB24 is a transcription factor which binds the promoter region of CDCA7 and is required for its expression (*Wu et al., 2016*). As CDCA7 binds HELLS to form the CHIRRC, we proposed that its ATP-dependent nucleosome sliding activity exposes DNA that was previously wrapped around the histone octamer and makes it accessible for DNA methylation (*Jenness et al., 2018*). Indeed, DNMT3A and DNMT3B cannot methylate DNA within a nucleosome (*Felle et al., 2011*), and the importance of HELLS and DDM1 for DNA methylation at nucleosomal DNA has been reported in mouse embryonic fibroblasts and *Arabidopsis*, respectively (*Lyons and Zilberman, 2017*). Given the frequent loss of 5mC as an epigenetic mark in multiple evolutionary lineages, it is striking that the role for HELLS/DDM1 in DNA methylation is conserved in evolutionarily distant mammals and plants. Importantly, this would suggest that the promotion of DNA methylation through nucleosome sliding is specific to HELLS and cannot be substituted by other SNF2 family nucleosome remodelers, such as SNF2 (SMARCA2/4), INO80, and ISWI (SMARCA1/5). If the specific function of HELLS and CDCA7 in DNA methylation is indeed derived from the last eukaryotic common ancestor (LECA), we hypothesize that HELLS and CDCA7 coevolved with other DNA methylation machineries. We here test this hypothesis and discuss the potential sequence of events that led to the loss of DNA methylation in some species.

## Results

### CDCA7 is absent from the classic model organisms that lack genomic 5mC

CDCA7 is characterized by the unique zf-4CXXC_R1 domain (Pfam PF10497 or Conserved Domain Database [CDD] cl20401) (*Lu et al., 2020*; *Mistry et al., 2021*). Conducting a BLAST search using human CDCA7 as a query sequence against the Genbank protein database, we realized that no zf-4CXXC_R1-containing proteins are identified in the classic model organisms *Drosophila melanogaster*, *Caenorhabditis elegans*, *Schizosaccharomyces pombe* and *Saccharomyces cerevisiae*, which are all also known to lack any DNMTs and genomic 5mC (*Zemach et al., 2010*). However, we identified a protein with a zf-4CXXC_R1 motif in the bumblebee *Bombus terrestris* and the thale cress *A. thaliana*, which have both maintenance and de novo DNMTs (*Bewick et al., 2017b*; *Chan et al., 2005*; *Li et al., 2018*; *Zemach et al., 2010*; *Figure 1* and *Figure 1—source data 1*). Based on the reciprocal best hits (RBH) criterion using human HELLS as a query sequence (see Methods) (*Ward et al., 2014*), HELLS-like proteins were identified in *A. thaliana* (DDM1) and *S. cerevisiae* (Irc5) in line with previous reports (*Litwin et al., 2017*), as well as in *B. terrestris*. No clear HELLS-like proteins were identified in *D. melanogaster*, *C. elegans*, or *S. pombe*. This pilot-scale analysis based on the RBH criterion led us to hypothesize that the evolutionary maintenance of CDCA7 is linked to that of HELLS and DNMTs. In

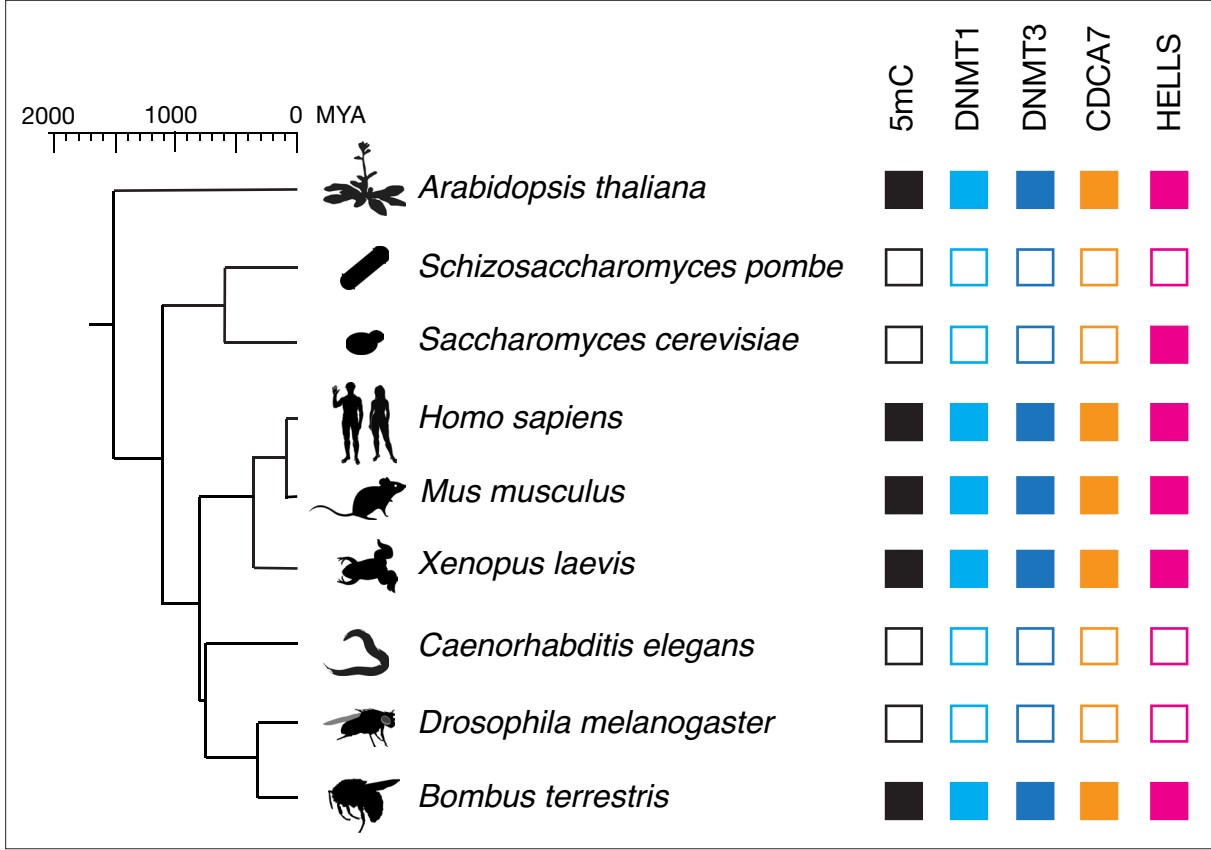

**Figure 1.** CDCA7 is absent from model organisms with undetectable genomic 5mC. Filled squares and open squares indicate presence and absence of an orthologous protein(s), respectively. CDCA7 homologs are absent from model organisms where DNMT1, DNMT3 and 5mC on genomic are absent.

The online version of this article includes the following source data for figure 1:

**Source data 1.** Lists of proteins and species used in this study.

order to statistically validate this hypothesis, we set out to systematically identify and classify homologs of CDCA7, HELLS and DNMTs in a broad range of evolutionary lineages.

## CDCA7 family proteins in vertebrates

We first characterized evolutionary conservation of CDCA7 family proteins in vertebrates, where 5mC, DNMT1, and DNMT3 are highly conserved. A BLAST search against the Genbank protein database identified two zf-4CXXC_R1 domain-containing proteins, CDCA7/JPO1 and CDCA7L/R1/JPO2, throughout Gnathostomata (jawed vertebrates; *Figure 2A*, *Figure 2B*, *Figure 2—figure supplement 1*, and *Figure 1—source data 1*). In frogs (such as *Xenopus*, but not all amphibians), and some fishes (such as *Astyanax mexicanus* and *Takifugu rubripes*), a third paralog CDCA7e exists. CDCA7e is the only CDCA7-like protein that can be detected in *Xenopus* eggs (*Jenness et al., 2018*), and thus likely represents a form specific to oocytes and early embryos in these species. Among twelve conserved cysteine residues originally reported in the zf-4CXXC_R1 domain (*Ou et al., 2006*), the 12th cysteine residue is not conserved in *Rhincodon typus* (whale shark) CDCA7 and in *Xenopus laevis* CDCA7e. In general, the 12th cysteine residue of the zf-4CXXC_R1 domain is least conserved among CDCA7 homologs within and outside of vertebrates, such that we do not consider it a key component of the zf-4CXXC_R1 domain (see below). Considering that the jawless fish *Petromyzon marinus* (sea lamprey) and other invertebrates commonly possess only one CDCA7 family gene (*Figure 2*, see Figure 6 and *Figure 1—source data 1*), the CDCA7 paralogs may have emerged in jawed vertebrates. Overall, the zf-4CXXC_R1 sequence is highly conserved among vertebrate CDCA7 homologs, including three residues that are mutated in ICF3 patients (R274, G294 and R304 in human CDCA7) (*Figure 2B*; *Thijssen et al., 2015*). Note that these amino acid positions of the ICF3 mutations in human CDCA7 are based

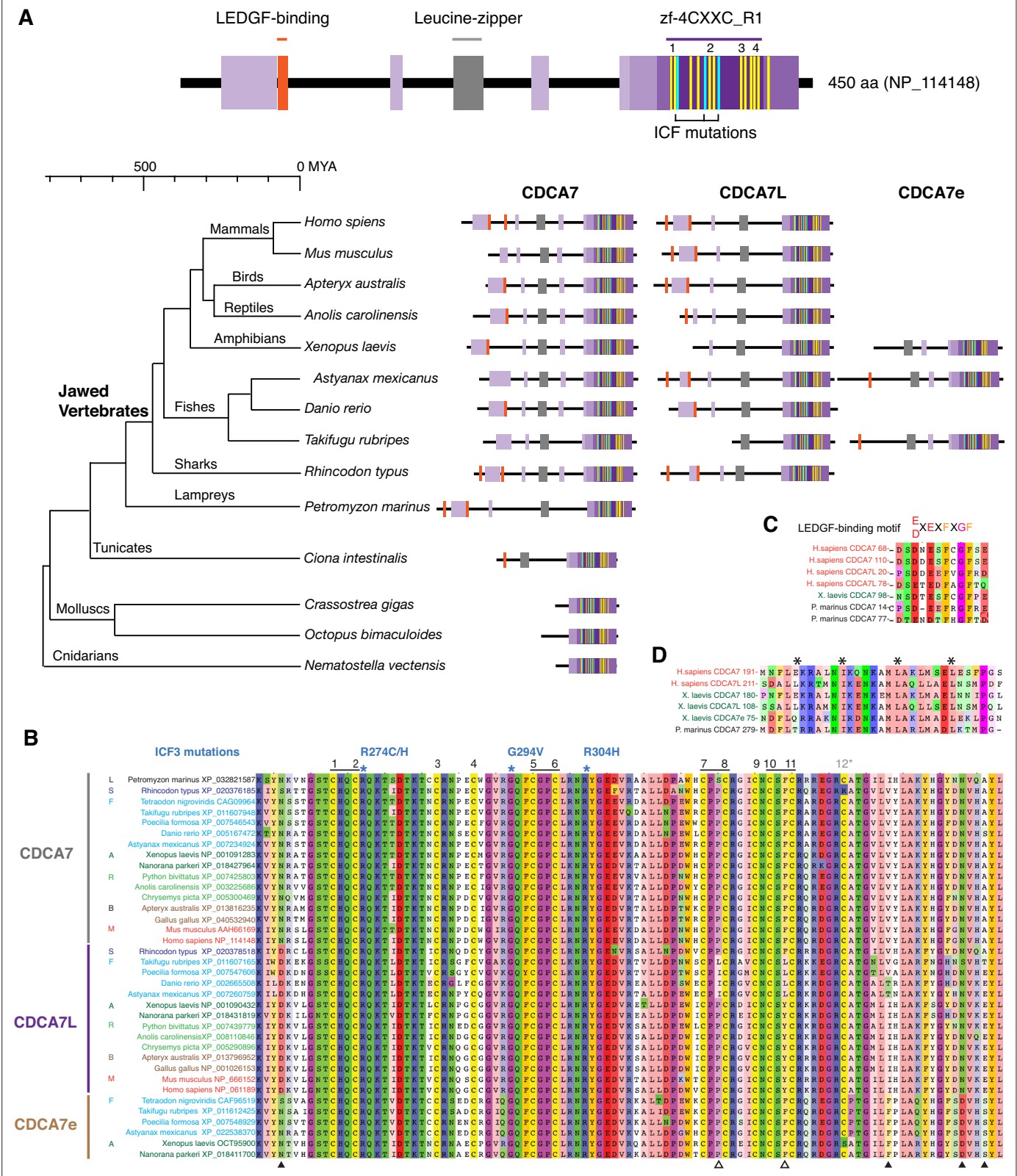

**Figure 2.** CDCA7 paralogs in vertebrates. (**A**) Schematics of vertebrate CDCA7 primary sequence composition, based on NP_114148. Yellow lines and light blue lines indicate positions of evolutionary conserved cysteine residues and residues that are mutated in ICF patients, respectively. (**B**) Sequence alignment of the zf-4CXXC_R1 domain of vertebrate CDCA7-family proteins. White arrowheads; amino residues unique in fish CDCA7L. Black

*Figure 2 continued on next page*

*Figure 2 continued*

arrowheads; residues that distinguish CDCA7L and CDCA7e from CDCA7. (**C**) Sequence alignment of LEDGF-binding motifs. (**D**) Sequence alignment of the conserved leucine-zipper.

The online version of this article includes the following source data and figure supplement(s) for figure 2:

**Source data 1.** Multiple sequence alignment of zf-4CXXC_R1 domains.

**Source data 2.** An IQ-TREE result of the consensus phylogenetic tree generation of zf-4CXXC_R1 containing proteins.

**Figure supplement 1.** Evolutionary conservation of CDCA7-family proteins and other zf-4CXXC_R1-containig proteins.

on the previously reported sequence of an NP_665809 (*Thijssen et al., 2015*), which is annotated as isoform 2 (371 amino acids), whereas we list the isoform 1 (NP_114148) with 450 amino acids in this study (*Figure 2A*).

While the presence of four CXXC motifs in the zf-4CXXC_R1 domain is reminiscent of a classic zinc finger-CXXC domain (zf-CXXC, Pfam PF02008), which binds to nonmethylated CpG (*Long et al., 2013*), their cysteine arrangement is distinctly different, perhaps reflecting the capacity of zf-4CXXC_R1 domain to recognize nucleosomes (*Jenness et al., 2018*) and potentially specific epigenetic marks or DNA sequences. In vertebrate CDCA7 paralogs, 11 conserved cysteines are arranged as $\underline{CXXC}X_{10}CX\text{-}_4CX_7\underline{CXXC}X_{19}\underline{CXXC}X_3CX\underline{CXXC}$. In contrast, in the classic zf-CXXC domain eight cysteines are arranged as $\underline{CXXCXXC}X_{4\text{-}5}\underline{CXXCXXC}X_{8\text{-}14}CX_4C$ (*Long et al., 2013*). Apart from the zf-4CXXC_R1 domain, vertebrate CDCA7 family proteins often, but not always, contain one or two Lens epithelium-derived growth factor (LEDGF)-binding motif(s) (*Figure 2A* and *Figure 2C*), defined as ([E/D]XEXFXGF; *Tesina et al., 2015*). It has been reported that human CDCA7L and CDCA7 both interact with c-Myc but

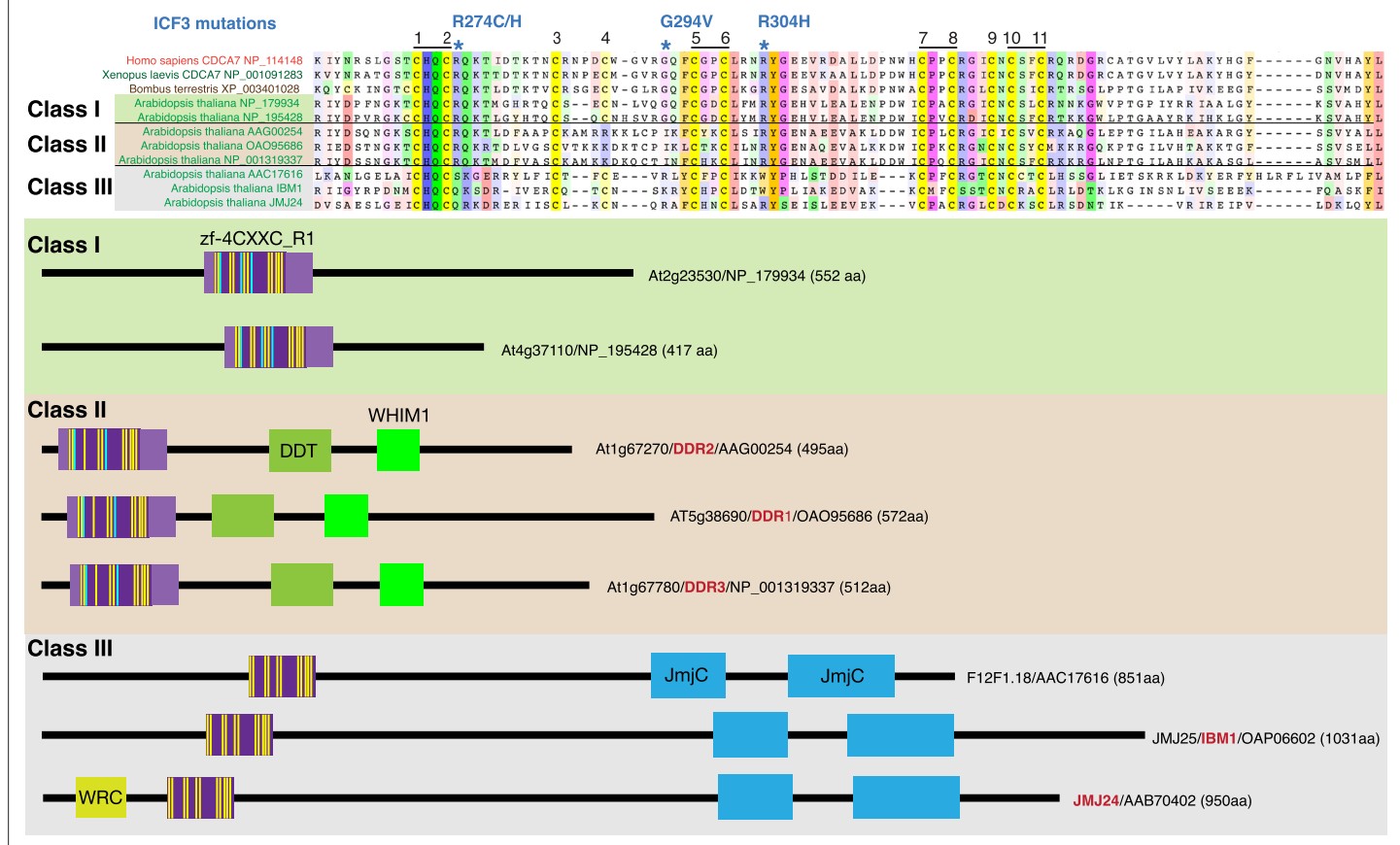

**Figure 3.** CDCA7 homologs and other zf-4CXXC_R1-containing proteins in *Arabidopsis*. Top; alignments of the zf-4CXXC_R1 domain found in *Arabidopsis thaliana*. Bottom; domain structure of the three classes of zf-4CXXC_R1-containing proteins in *Arabidopsis*.

The online version of this article includes the following figure supplement(s) for figure 3:

**Figure supplement 1.** Sequence alignment and classification of zf-4CXXC_R1 domains across eukaryotes.

apparently via different regions, a leucine zipper and a less-defined adjacent segment that overlaps with a bipartite nuclear localization signal, respectively (*Gill et al., 2013*; *Huang et al., 2005*). This leucine zipper sequence is highly conserved among vertebrate CDCA7 family proteins (*Figure 2A* and *Figure 2D*). In contrast to zf-CXXC domain-containing proteins such as KDM2A/B, DNMT1, MLL1/2, and TET1/3, the vertebrate CDCA7 family proteins do not contain any predicted enzymatic domains (*Huang et al., 2005*; *Maertens et al., 2006*; *Tesina et al., 2015*).

## Plant homologs of CDCA7

A BLAST sequence homology search identified three classes of zf-4CXXC_R1 domain-containing proteins in *Arabidopsis thaliana* (*Figure 3*). Class I proteins have one zf-4CXXC_R1 domain with no other detectable domains listed in CDD or Pfam/InterPro (*Paysan-Lafosse et al., 2023*). Strikingly, all three ICF-associated residues (R274, G294, and R304 in human CDCA7) as well as all eleven characteristic cysteines of the zf-4CXXC_R1 domain are conserved, with the exception that the position of the fourth cysteine is shifted two residues toward the N terminus. We define the domain with the conservation of the signature cysteine residues and three ICF-associated residues as 'class I zf-4CXXC_R1' and hypothesize that proteins with class I zf-4CXXC_R1 are CDCA7 orthologs (including their highly homologous paralogs in this report).

Class II proteins contain a zf-4CXXC_R1 domain, a DDT domain and a WHIM1 domain. These proteins were previously identified as DDR1-3 (*Dong et al., 2013*). DDT and WHIM1 domains are commonly found in proteins that interact with SNF2h/ISWI (*Aravind and Iyer, 2012*; *Li et al., 2017*; *Yamada et al., 2011*). Indeed, it was reported that *Arabidopsis* DDR1 and DDR3 interact with the ISWI orthologs CHR11 and CHR17 (*Tan et al., 2020*). Among the eleven cysteine residues in the zf-4CXXC_R1 domain of these proteins, the position of the fourth residue is shifted towards the C-terminus. The ICF-associated glycine residue (G294 in human CDCA7, mutated to valine in ICF3 patients) is replaced by isoleucine. We define the zf-4CXXC_R1 containing a substitution at this glycine residue as class II zf-4CXXC_R1.

Class III proteins are longer (~1000 amino acid) and contain an N-terminal zf-4CXXC_R1 domain and a C-terminal JmjC domain (Pfam, PF02373), which is predicted to possess demethylase activity against histone H3K9me2/3 (*Saze et al., 2008*). While all 11 cysteine residues can be identified, there are deletions between the 4th and 5th cysteine and 6th and 7th cysteine residues. None of the ICF-associated residues are conserved in the class III. One of these class III proteins is IBM1 (increase in bonsai mutation 1), whose mutation causes the dwarf "bonsai" phenotype (*Saze et al., 2008*), which is accompanied with increased H3K9me2 and DNA methylation levels at the *BONSAI* (*APC13*) locus. Double mutants of *ddm1* and *ibm1* exacerbate the bonsai phenotype, indicating that DDM1 and IBM1 act independently to regulate DNA methylation (*Saze et al., 2008*). Another class III protein is JMJ24, which harbors a RING finger domain in addition to the 4CXXC and JmjC domains. This RING finger domain promotes ubiquitin-mediated degradation of the DNA methyltransferase CMT3, and thus opposes DNA methylation (*Deng et al., 2016*).

Homologs of these three classes of CDCA7 proteins found in *Arabidopsis* are widely identified in green plants (Viridiplantae), including Streptophyta (e.g. rice, maize, moss, fern) and Chlorophyta (green algae; *Figure 3—figure supplement 1* and *Figure 1—source data 1*). Other variants of zf-4CXXC_R1 are also found in Viridiplantae. In contrast to green plants, in which the combined presence of HELLS/DDM1-, CDCA7- and DNMT- orthologs is broadly conserved, no zf-4CXXC_R1-containing proteins can be identified in red algae (Rhodophyta; *Figure 1—source data 1*, see Figure 5 and *Figure 5—figure supplement 1*).

## Zf-4CXXC_R1-containing proteins in Fungi

Although *S. pombe* and *S. cerevisiae* genomes do not encode any CDCA7 family proteins, a BLAST search identified various fungal protein(s) with a zf-4CXXC_R1 domain. Among the zf-4CXXC_R1-containing proteins in fungi, 10 species (*Kwoniella mangroviensis*, *Coprinopsis cinere*, *Agaricus bisporus*, *Taphrina deformans*, *Gonapodya prolifera*, *Basidiobolus meristosporus*, *Coemansia reversa*, *Linderina pennispora*, *Rhizophagus irregularis*, *Podila verticillate*) harbor a class II zf-4CXXC_R1 domain with two notable deviations (*Figure 4A*). First, the space between the third and fourth cysteine residues is variable. Second, the fifth cysteine is replaced by aspartate in Zoopagomycota (*Basidiobolus meristosporus* ORX82853, *Coemansia reversa* PIA14937, *Linderina pennispora* ORX71196) (*Figure 4*). As

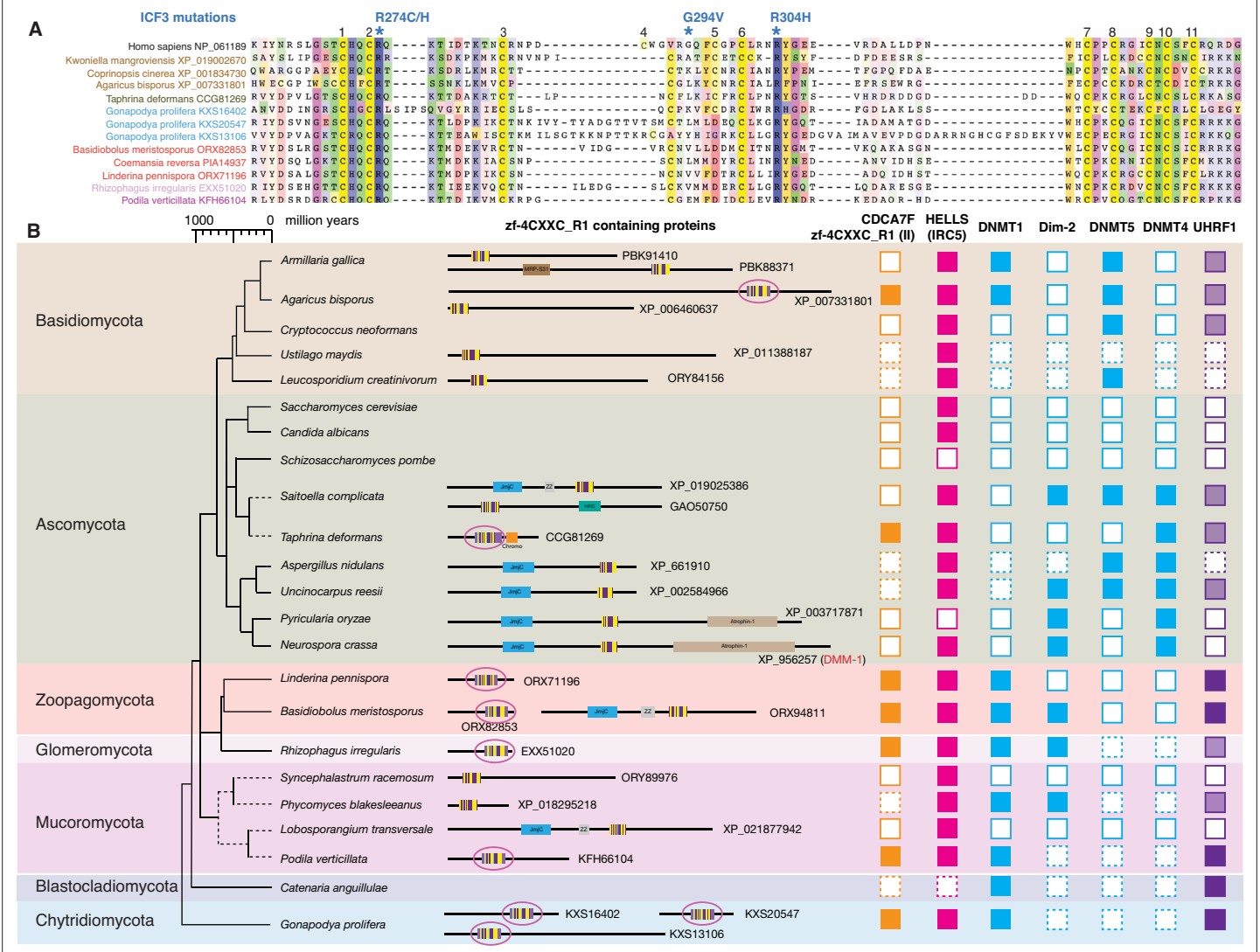

**Figure 4.** Evolutionary conservation of CDCA7F, HELLS and DNMTs in fungi. (**A**) Sequence alignment of fungi-specific CDCA7F with class II zf-4CXXC_R1 sequences. (**B**) Domain architectures of zf-4CXXC_R1-containing proteins in fungi. The class II zf-4CXXC_R1 domain is indicated with purple circles. Squares with dotted lines indicate preliminary genome assemblies. Opaque boxes of UHRF1 indicate homologs that harbor the SRA domain but not the RING-finger domain.

these proteins do not contain any additional CDD-annotated domain like vertebrate and plant CDCA7 orthologs (except for *Taphrina deformans* CCG81269, which also has a CHROMO domain), we define this fungal protein family as CDCA7F, which forms a distinct clade in our phylogenetic analysis of zf-4CXXC_R1 sequence alignment (*Figure 2—figure supplement 1* and *Figure 3—figure supplement 1*).

Besides CDCA7F, several fungal species encode a protein with a diverged zf-4CXXC_R1 domain, including those with a JmjC domain at the N-terminus (*Figure 4B*, *Figure 1—source data 1*). This composition mimics the plant class III proteins, for which the JmjC domain is located at the C-terminus. Among these proteins, it was suggested that *Neurospora crassa* DMM-1 does not directly regulate DNA methylation or demethylation but rather controls the deposition of histone H2A.Z and/ or H3K56 acetylation, which inhibit spreading of heterochromatin segments with methylated DNA and H3K9me3 (*Honda et al., 2010*; *Zhang et al., 2022*). Thus, the known roles of these divergent variants of zf-4CXXC_R1 are not directly associated with DNA methylation.

## Systematic identification of CDCA7 and HELLS homologs in eukaryotes

To systematically identify CDCA7 and HELLS homologs in the major eukaryotic supergroups, we conducted a BLAST search against the NCBI protein database using human CDCA7 and HELLS protein sequences. To omit species with a high risk of false negative identification, we selected species containing at least 6 distinct proteins with compelling homology to the SNF2 ATPase domain of HELLS, based on the assumption that each eukaryotic species is expected to have 6–20 SNF2 family ATPases (*Flaus et al., 2006*). Indeed, even the microsporidial pathogen *Encephalitozoon cuniculi*, whose genome size is a mere 2.9 Mb, contains six SNF2 family ATPases (*Flaus et al., 2006*). As such, we generated a panel of 180 species encompassing all major eukaryote supergroups (5 Excavata, 18 SAR [2 Rhizaria, 6 Alveolata, 10 Stramenopiles]), 1 Haptista, 1 Cryptista, 15 Archaeplastida [3 Rhodophyta and 12 Viridiplantae], 4 Amoebozoa, 136 Opisthokonts [34 Fungi, 3 Holozoa, and 99 Metazoa] (*Figure 5—figure supplement 1*, and *Figure 1—source data 1*).

To annotate HELLS orthologs, a phylogenetic tree was constructed from a multiple sequence alignment of the HELLS homologs identified based on the RBH criterion alongside other SNF2-family proteins of *H. sapiens*, *D. melanogaster, S. cerevisiae*, and *A. thaliana*. If HELLS orthologs are correctly identified (i.e. without erroneously including orthologs of another SNF2-subfamily) they should cluster together in a single clade. However, the sequence alignment using the full-length protein sequence failed to cluster HELLS and DDM1 in the same clade (*Figure 5—figure supplement 2*). Since HELLS and other SNF2-family proteins have variable insertions within the SNF2 ATPase domain, multiple sequence alignment of the SNF2 domains was then conducted after removing the insertion regions, as previously reported (*Flaus et al., 2006*). By this SNF2 domain-only alignment method, all HELLS orthologs formed a clade, separated from CHD1, ISWI, SMARCA2/4, SRCAP, and INO80 (*Figure 5—figure supplement 3*). An independent phylogenetic tree construction with a maximum-likelihood based method and 1000 bootstrap replicates confirmed these assignments (see Methods, *Figure 5—source data 1* and *Figure 5—source data 2*).

A BLAST search with the human CDCA7 sequence across the panel of 180 species identified a variety of proteins containing the zf-4CXXC_R1 domain, which is prevalent in all major supergroups (*Figure 5*, *Figure 5—figure supplement 1*, and *Figure 1—source data 1*). Each of these identified proteins contains only one zf-4CXXC_R1 domain. The resulting list of CDCA7 BLAST hits were further classified as prototypical CDCA7 orthologs if they preserve the criteria of the class I zf-4CXXC_R1 (signature 11 cysteine residues *and* the three ICF-associated residues) (*Figure 3—figure supplement 1*). A phylogenetic tree analysis of zf-4CXXC_R1 domains from diverse species confirmed that these CDCA7 orthologs with the class I zf-4CXXC_R1 domain are clustered under the same clade (*Figure 2—figure supplement 1*, also see Methods, *Figure 2—source data 1* and *Figure 2—source data 2*). CDCA7 orthologs are broadly found in the three supergroup lineages (Archaeplastida, Amoebozoa, Opisthokonta; *Figure 5*, *Figure 5—figure supplement 1*, and *Figure 1—source data 1*). In Excavata, the amoeboflagellate *Naegleria gruberi* encodes a protein that is a likely ortholog of CDCA7 with an apparent C-terminal truncation (XP_002678720), possibly due to a sequencing error (*Figure 3—figure supplement 1*). In SAR, CDCA7 orthologs are absent from all available genomes, except stramenopile *Tribonema minus*, which encodes a distantly related likely ortholog of CDCA7 (KAG5177154). These conservations suggest that the class I zf-4CXXC_R1 domain in CDCA7 was inherited from the LECA.

In addition to the class I zf-4CXXC_R1 domain, we also identified divergent zf-4CXXC_R1 domains across eukaryotes, although metazoan species only contain CDCA7 orthologs (and their paralogs such as CDCA7L and CDCA7e) with the exception of the sponge *Amphimedon queenslandica*, which also encodes a protein (XP_019849176) with the class II zf-4CXXC_R1 domain, DDT, and WHIM1 domain, the composition of which is similar to plant DDR1-3 (*Figure 3*, *Figure 5*, *Figure 3—figure supplement 1*, and *Figure 1—source data 1*). The Amoebozoa *Acanthamoeba castellanii* encodes a protein (XP_004340890) containing the class II zf-4CXXC_R1 domain as well as the CHROMO domain (cd00024) (*Figure 3—figure supplement 1* and *Figure 1—source data 1*). This domain combination is also found in the *Taphrina deformans* CDCA7F, CCG81269 (*Figure 4*). Proteins with a combinatory presence of a diverged zf-4CXXC_R1 domain and the JmjC domain are found in plants, fungi, Amoebozoa, and *Naegleria gruberi* (*Figure 1—source data 1*).

Despite the prevalence of the zf-4CXXC_R1 domain and its variants in eukaryotes, no zf-4CXXC_R1 domain was found in prokaryotes and Archaea. This is in contrast to SNF2 family proteins and DNA

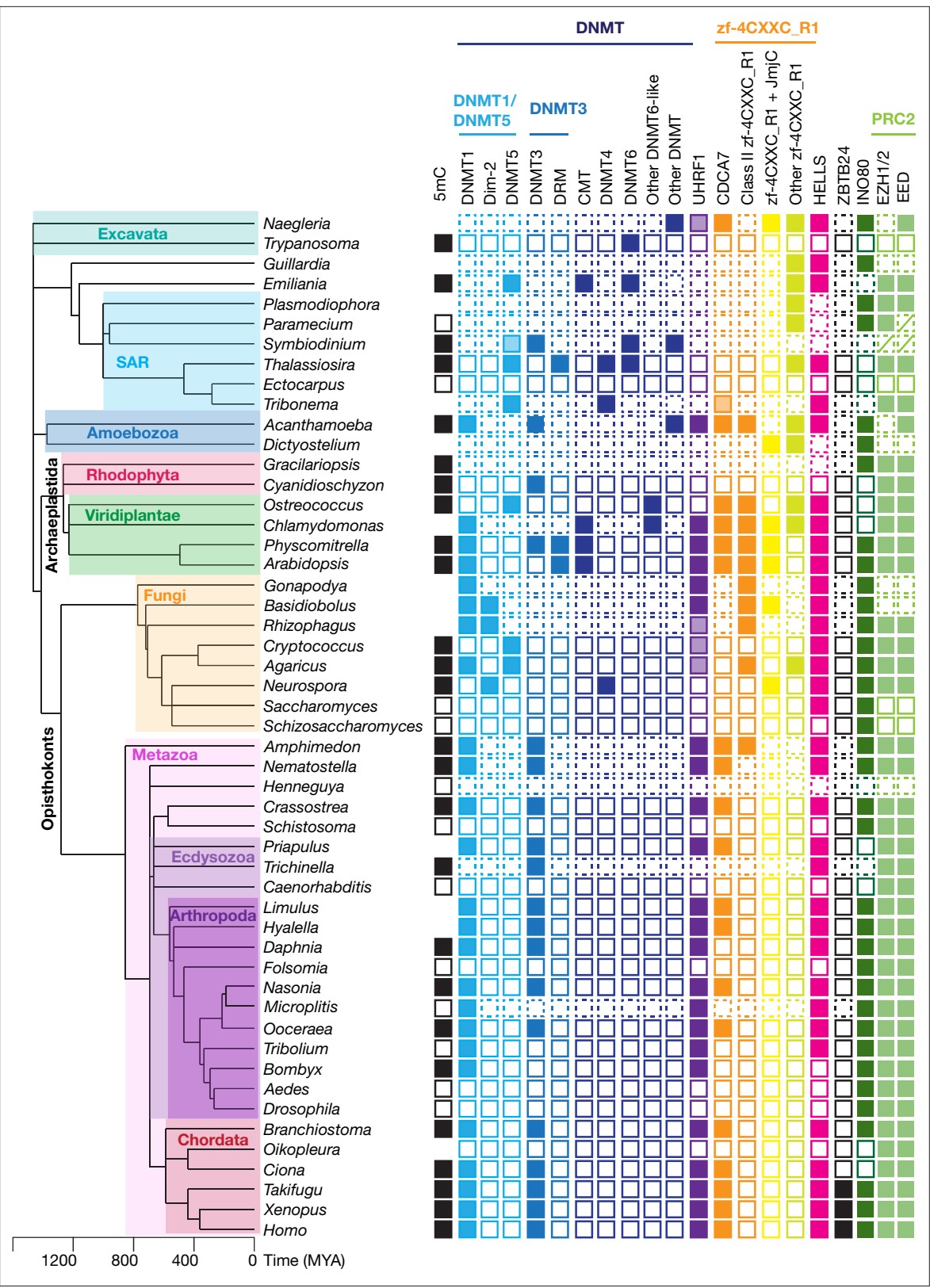

**Figure 5.** Evolutionary conservation of CDCA7, HELLS, and DNMTs. The phylogenetic tree was generated based on Timetree 5 (*Kumar et al., 2022*). Filled squares and open squares indicate presence and absence of an orthologous protein(s), respectively. Squares with dotted lines imply preliminary-level genome assemblies. Squares with a diagonal line; *Paramecium* EED was functionally identified (*Miró-Pina et al., 2022*), but not by the sequence-based search in this study; homologs of EZH1/2 and EED were identified in *Symbiodinium sp. KB8* but not in *Symbiodinium microadriaticum* (*Figure*

*Figure 5 continued on next page*

*Figure 5 continued*

*1—source data 1*). An opaque box of DNMT5 in *Symbiodinium* indicates a homolog that does not contain the ATPase domain, which is commonly found in DNMT5 family proteins. Opaque boxes of UHRF1 indicate homologs that harbor the SRA domain but not the RING-finger domain. Full set of analysis on the panel of 180 eukaryote species is shown in *Figure 5—figure supplement 1* and *Figure 1—source data 1*. Genbank accession numbers of each protein and PMID numbers of published papers that report presence or absence of 5mC are reported in *Figure 1—source data 1*.

The online version of this article includes the following source data and figure supplement(s) for figure 5:

**Source data 1.** Multiple sequence alignment of the SNF2 ATPase domains of HELLS homologs and other SNF2-family proteins.

**Source data 2.** An IQ-TREE result of the consensus phylogenetic tree generation of HELLS homologs and other SNF2-family proteins.

**Figure supplement 1.** Evolutionary conservation of CDCA7, HELLS, and DNMTs.

**Figure supplement 2.** Phylogenetic tree of HELLS and other SNF2 family proteins.

**Figure supplement 3.** Phylogenetic tree of the SNF2-domain.

**Figure supplement 4.** Phylogenetic tree of DNMT proteins.

**Figure supplement 4—source data 1.** Multiple sequence alignment of DNA methyltransferase domains for *Figure 5—figure supplement 4*. DNMT domains from various DNMTs were aligned by MUSCLE v5.

**Figure supplement 4—source data 2.** An IQ-TREE result of the consensus phylogenetic tree generation of DNMTs for *Figure 5—figure supplement 4*.

*Figure 5—figure supplement 4—source data 1* was used for the analysis by IQ-TREE.

methyltransferases, which can be identified in prokaryotes and Archaea (*Colot and Rossignol, 1999*; *Flaus et al., 2006*; *Huff and Zilberman, 2014*; *Ponger and Li, 2005*), pointing toward a possibility that the zf-4CXXC_R1 domain emerged to deal with unique requirement of eukaryotic chromatin.

## Classification of DNMTs in eukaryotes

A simple RBH approach is not practical to classify eukaryotic DNMT proteins due to the presence of diverse lineage-specific DNMTs (*Huff and Zilberman, 2014*). Therefore, we collected proteins with a DNMT domain within the panel of 180 eukaryote species, and then extracted the DNMT domains from each sequence (based on an NCBI conserved domains, Dcm [COG0270], Dcm super family [cl43082], AdoMet_MTases superfamily [cl17173]). Generating a phylogenetic tree based on the multi-sequence alignment of the DNMT domains, we were able to classify the majority of all identified DNMTs as previously characterized DNMT subtypes according to their sequence similarity (*Figure 5—figure supplement 4* and ). These DNMT subtypes include DNMT1; DNMT3; the plant specific de novo DNA methyltransferases DRM1-3; the 'true' plant DNMT3 orthologs *Yaari et al., 2019*; the plant-specific CMT *Bewick et al., 2017a*; the fungi-specific maintenance methyltransferase Dim-2 and de novo methyltransferase DNMT4 *Bewick et al., 2019a*; *Nai et al., 2020*; the SNF2 domain-containing maintenance methyltransferase DNMT5 *Dumesic et al., 2020*; *Huff and Zilberman, 2014*; DNMT6 (a poorly characterized putative DNMT identified in Stramenopiles, Haptista and Chlorophyta) (*Huff and Zilberman, 2014*), and the tRNA methyltransferase TRDMT1 (also known as DNMT2; *Figure 5—figure supplement 4*). In this report, we call a protein DNMT3 if it clusters into the clade including metazoan DNMT3, plant DNMT3, and DRM. We also identified other DNMTs, which did not cluster into these classes. For example, although it has been reported that DNMT6 is identified in *Micromonas* but not in other Chlorophyta species such as *Bathycoccus* and *Ostreococcus* (*Huff and Zilberman, 2014*), we identified Chlorophyta-specific DNMTs that form a distinct clade, which seems to be diverged from DNMT6 and DNMT3. We temporarily called this class Chlorophyta DNMT6-like (*Figure 5—figure supplement 4*). Other orphan DNMTs include the de novo DNA methyltransferase DNMTX in fungus *Kwoniella mangroviensis* (*Catania et al., 2020*), and an uncharacterized DNMT in *N. gruberi* (XP_002682263).

## Coevolution of CDCA7, HELLS and DNMTs

The classification of homologs of CDCA7, HELLS and DNMTs across the panel of 180 eukaryotic species reveals that they are conserved across the major eukaryote supergroups, but they are also dynamically lost (*Figure 5*, *Figure 5—figure supplement 1* and *Figure 1—source data 1*). We found 40 species encompassing Excavata, SAR, Amoebozoa, and Opisthokonta that lack CDCA7 (or CDCA7F), HELLS and DNMT1. Species that encode the set of DNMT1, UHRF1, CDCA7, and

HELLS are particularly enriched in Viridiplantae and Metazoa. A clear exception in Amoebozoa is *Acanthamoeba castellanii*, whose genome also encodes these four proteins and is reported to have methylated cytosines (*Moon et al., 2017*). *N. gruberi* is an exceptional example among Excavata species, encoding a putative CDCA7 (XP_002678720) as well as HELLS; an orphan DNMT protein (XP_002682263), and a UHRF1-like protein. DNMT1, DNMT3, HELLS, and CDCA7 seem to be absent in other Excavata lineages, although Euglenozoa variants of DNMT6 and cytosine methylation are identified in *Trypanosoma brucei* and *Leishmania major* (*Huff and Zilberman, 2014*; *Militello et al., 2008*). The yellow-green algae *Tribonema minus* is the only SAR species that encodes a putative CDCA7 (KAG5177154), along with DNMT5 and HELLS (*Figure 5*, *Figure 2—figure supplement 1*, and *Figure 3—figure supplement 1*).

Among the panel of 180 eukaryote species, we found 82 species that encode CDCA7 (including fungal CDCA7F) (*Figure 5—figure supplement 1* and *Figure 1—source data 1* tab6). Strikingly, all 82 species containing CDCA7 (or CDCA7F) also harbor HELLS. Almost all CDCA7 encoding species also possess DNMT1. Exceptions are: the yellow-green algae *T. minus* and the Mamiellophyceae lineage of the green algae/Chlorophyta (*Bathycoccus prasinos*, *Ostreococcus lucimarinus*, *Micromonas pusilla*), which lost DNMT1 but possess DNMT5; the amoeboflagellate *N. gruberi*, which encodes an orphan DNMT and UHRF1, and the fungus *T. deformans*, which encodes UHRF1 and DNMT4. In contrast, 20 species (e.g. *S. cerevisiae*) possess only HELLS, while 10 species (e.g. the silk moth *Bombyx mori*) retains only DNMT1 among the set of DNMT1, CDCA7 and HELLS. These observations indicate that the function of CDCA7 is strongly linked to HELLS and DNMT1, such that the presence of CDCA7 depends on HELLS and DNMT1, while CDCA7 is easier to lose than DNMT1 and HELLS. Compared to the apparent HELLS/DNMT1-dependent existence of CDCA7, DNMT3 seems to be more dispensable; besides Fungi, which lack DNMT3, 16 species (e.g., the bed bug *Cimex lectularius* and the red paper wasp *Polistes canadensis*) possess the set of CDCA7, HELLS, and DNMT1 but not DNMT3/DNMT3-like proteins (*Figure 5—figure supplement 1* and *Figure 1—source data 1* tab6).

To quantitatively assess coevolution of DNMTs, CDCA7 and HELLS, we performed CoPAP analysis on the panel of 180 eukaryote species (*Figure 6—figure supplement 1*; *Cohen et al., 2013*). The analysis was complicated due to the lineage-specific diverse DNMT classes (e.g. Dim2, DNMT5, DNMT6 and other plant specific DNMT variants) and divergent variants of zf-4CXXC_R1. Considering this caveat, we conducted CoPAP analysis of five DNMTs (DNMT1, Dim-2, DNMT3 [including DRM], DNMT5, DNMT6), UHRF1, HELLS, CDCA7, proteins with class II zf-4CXXC_R1, and proteins with zf-4CXXC_R1 and JmjC. As fungi-specific CDCA7F contains class II zf-4CXXC_R1, and all the other class II zf-4CXXC_R1 containing proteins were identified in species that also possess CDCA7, we conducted CoPAP against two separate lists; in the first list (*Figure 6—figure supplement 1A*) CDCA7F was included in the CDCA7 category (i.e. considered as a prototypical CDCA7 ortholog in fungi), whereas in the second list (*Figure 6—figure supplement 1B*) CDCA7F was included in the class II zf-4CXXC_R1 category. As positive and negative controls for the CoPAP analysis, we also included subunits of the PRC2 complex (EZH1/2, EED and Suz12), and other SNF2 family proteins SMARCA2/SMARCA4, INO80 and RAD54L, which have no direct role related to DNA methylation, respectively. As expected for proteins that act in concert within the same biological pathway, both CoPAP results showed significant coevolution between DNMT1 and UHRF1, as well as between the PRC2 subunits EZH1 and EED. Suz12 did not show a significant linkage to other PRC2 subunits by this analysis, most likely due to a failure in identifying diverged Suz12 orthologs, such as those in *Neurospora* and *Paramecium* (*Jamieson et al., 2013*; *Miró-Pina et al., 2022*). The evolutionary linkage between DNMT5 and DNMT6 were also observed (*Figure 6—figure supplement 1*). In the result of CoPAP analysis against the first list, where CDCA7F was included in the CDCA7 category, CDCA7 exhibited a coevolutionary linkage to DNMT1 (*Figure 6—figure supplement 1A*). In the second CoPAP, where CDCA7F was included in the class II zf-4CXXC_R1 category, CDCA7 exhibited linkage to UHRF1 and HELLS (*Figure 6—figure supplement 1B*). Notably, none of these proteins show an evolutionary association with the PRC2 proteins or other SNF2 family proteins, while the coevolutionary linkage between CDCA7 and the DNMT1-UHRF1 cluster was reproducible.

We next conducted the CoPAP analysis against a panel of 50 Ecdysozoa species, where DNA methylation system is dynamically lost in multiple lineages (*Figure 6A*; *Bewick et al., 2017b*; *Engelhardt et al., 2022*), yet the annotation of DNMTs, UHRF1, CDCA7 and HELLS is unambiguous. As a negative control, we included INO80, which is dynamically lost in several Ecdysozoa lineages, such as

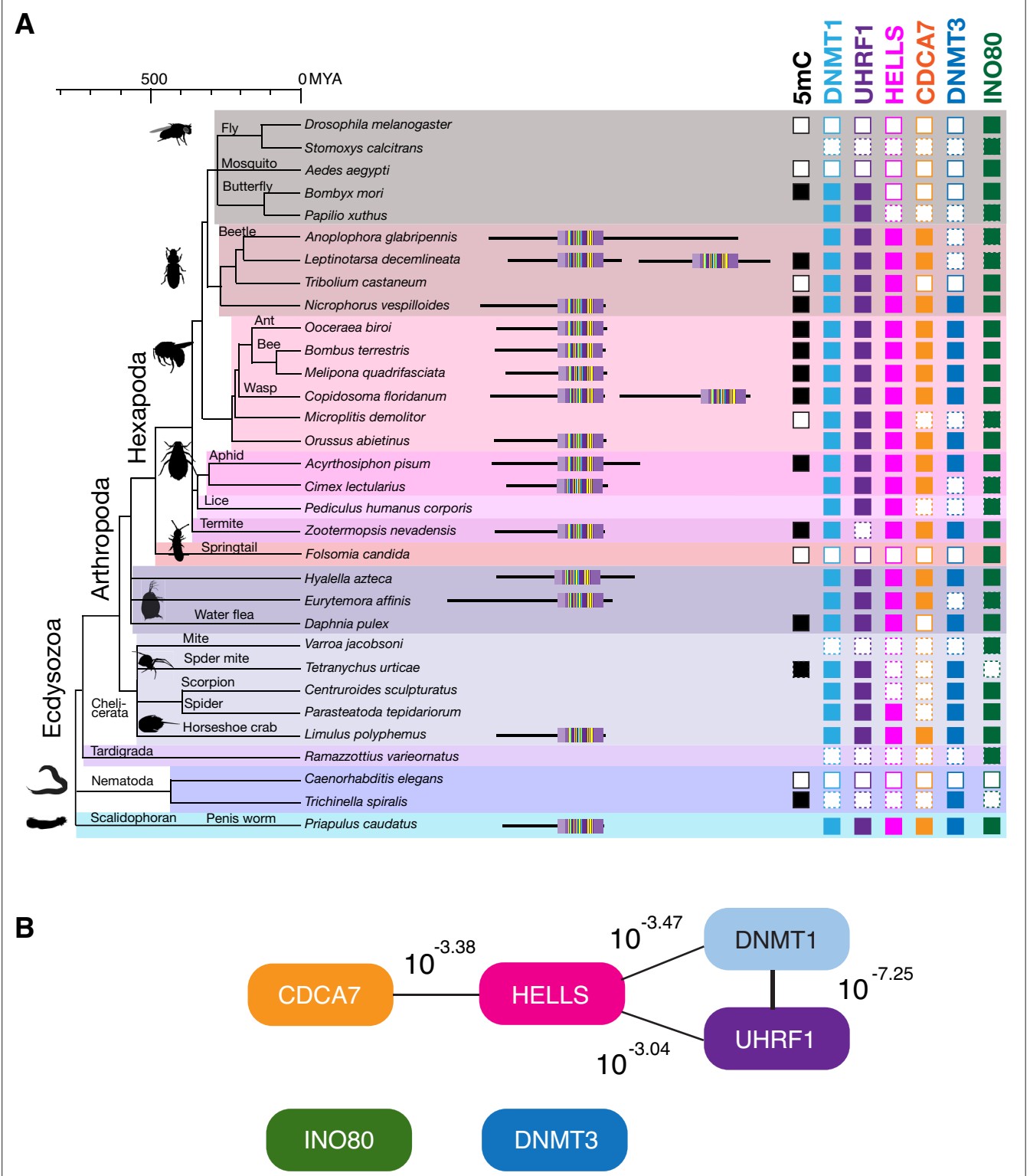

**Figure 6.** Coevolution of CDCA7, HELLS, UHRF1, and DNMT1 in Ecdysozoa. (**A**) Presence (filled squares) /absence (open squares) patterns of indicated proteins and genomic 5mC in selected Ecdysozoa species. Squares with dotted lines imply preliminary-level genome assemblies. Domain architectures of CDCA7 proteins with a zf-4CXXC_R1 domain are also shown. (**B**) CoPAP analysis of 50 Ecdysozoa species. Presence/absence patterns of indicated proteins during evolution were analyzed. List of species are shown in *Figure 1—source data 1*. Phylogenetic tree was generated by amino acid sequences of all proteins shown in *Figure 1—source data 1*. The number indicates the p-values.

The online version of this article includes the following figure supplement(s) for figure 6:

*Figure 6 continued on next page*

*Figure 6 continued*

**Figure supplement 1.** CoPAP analysis of CDCA7, HELLS, and DNMTs in eukaryotes.

---

*C. elegans* (*Figure 5*, *Figure 6A* and *Figure 5—figure supplement 1*). As expected, CoPAP analysis showed a highly significant coevolutionary interaction between DNMT1 and UHRF1 (*Figure 6B*). In addition, HELLS interacts with DNMT1, UHRF1 and CDCA7. In contrast, no linkage from INO80 or DNMT3 was determined. Altogether, these CoPAP analyses reproducibly found coevolutionary linkages between DNMT1-UHRF1 and CDCA7-HELLS, although the exact interaction topology is sensitive to the selection of species and protein classification methods.

## Loss of CDCA7 in braconid wasps together with DNMT1 or DNMT3

CoPAP analysis detected the coevolutionary linkage between CDCA7 and DNMT1-UHRF1, rather than DNMT3. We were therefore intrigued by the apparent absence of CDCA7 and DNMT3 in two insect species, the red flour beetle *Tribolium castaneum* and the braconid wasp *Microplitis demolitor*, whose genomic DNA does not have any detectable 5mC despite the presence of DNMT1 (*Bewick et al., 2017b*; *Schulz et al., 2018*; *Zemach et al., 2010*; *Figure 5*). To further validate the co-loss of CDCA7 and DNMT1/DNMT3 in insects, we focused on the Hymenoptera clade (including *M. demolitor*), for which genome synteny has been reported (*Li et al., 2021*). Indeed, a striking synteny is observed in the genome region surrounding CDCA7 among the parasitic wood wasp (*Orussus abietinus*) and Aculeata species (bees *Bombus terrestris* and *Habropoda laborlosa*, and the eusocial wasp *Polistes canadensis*), which diverged ~250 MYA (*Li et al., 2021*; *Peters et al., 2017*; *Figure 7* and *Figure 7—source data 1*). In these species, CDCA7 is located between Methyltransferase-like protein 25 homolog (MET25, E) and Ornithine aminotransferase homolog (OAT, F). In fact, the gene cluster containing LTO1 homolog (D), MET25 (E), OAT (F), and Zinc finger protein 808 (ZN808, G) is highly conserved in all the analyzed Hymenoptera species, but not outside of Hymenoptera (e.g. *Drosophila*).

However, CDCA7 is absent from this gene cluster in parasitoid wasps, including Ichneumonoidea wasps (braconid wasps [*M. demolitor*, *Cotesia glomerata*, *Aphidius gifuensis*, *Fopius arisanus*] and ichneumon wasp [*Venturia canescent*]) and chalcid wasps (*Copidosoma floridanum*, *Nasonia vitripennis*) (*Figure 7* and *Figure 7—source data 2*). Among Ichneumonoidea wasps, CDCA7 is undetectable in the genome of braconid wasps, while CDCA7 is located on a different chromosome in the ichneumon wasp *Venturia canescens*. In chalcid wasps, CDCA7 is present at a genome segment between Artemis (I) and Chromatin accessibility complex protein 1 (CHRC1, J). Supported by the chromosome-level genome assembly for *Cotesia glomerata* and *Aphidius gifuensis* (*Feng et al., 2020*; *Pinto et al., 2021*), the synteny analysis strongly indicates that CDCA7 is lost from braconid wasps. Intriguingly, braconid wasps co-lost CDCA7 either with DNMT3 (in *M. demolitor*, *Cotesia glomerata*, *Cotesia typhae* and *Chelonus insularis*) or with DNMT1 and UHRF1 (in *Fopius arisanus*, *Diachasma alloeum* and *Aphidius gifuensis*) (*Figure 7* and *Figure 7—source data 2*). Notably, in addition to *M. demolitor* (which lacks CDCA7 and DNMT3), it has been reported that little or no 5mC can be detected in *Aphidius ervi* (which lacks CDCA7, DNMT1 and UHRF1, *Figure 7—source data 2*; *Bewick et al., 2017b*). In contrast, all Hymenoptera species that are known to have detectable 5mC possess CDCA7 along with DNMT1, UHRF1, DNMT3, and HELLS, except for *Polistes canadensis*, which has all but DNMT3 (*Figure 7* and *Figure 7—source data 2*). This co-loss of CDCA7 and DNA methylation (together with either DNMT1-UHRF1or DNMT3) in braconid wasps suggests that evolutionary preservation of CDCA7 is more sensitive to DNA methylation status per se than to the presence or absence of a particular DNMT subtype. Consistent with this idea, among the panel of 180 eukaryote species, none of the 17 species where absence of genomic 5mC has been experimentally shown encodes CDCA7 (*Figure 1—source data 1* tab8) (*Aliaga et al., 2019*; *Antequera et al., 1984*; *Bewick et al., 2019a*; *Bewick et al., 2017b*; *Cock et al., 2010*; *Engelhardt et al., 2022*; *Geyer et al., 2011*; *Mondo et al., 2017*; *Noordhoek et al., 2018*; *Schulz et al., 2018*; *Singh et al., 2018*; *Zemach et al., 2010*).

## Discussion

Although DNA methylation is prevalent across eukaryotes, DNA methyltransferases are missing from a variety of lineages. Our study reveals that the nucleosome remodeling complex CHIRRC, composed of CDCA7 and HELLS, is frequently lost in conjunction with DNA methylation status. More specifically,

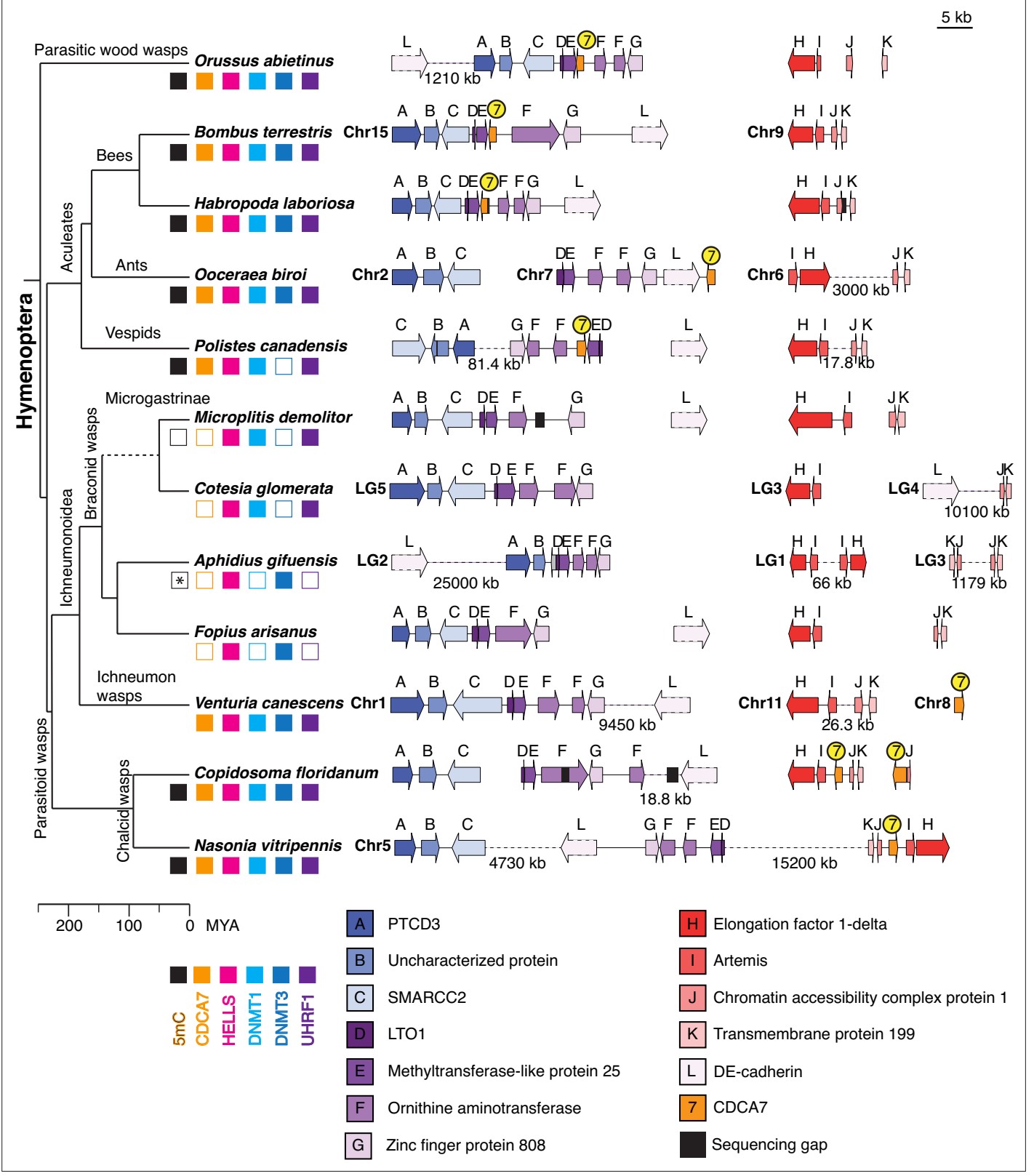

**Figure 7.** Synteny of Hymenoptera genomes adjacent to CDCA7 genes. Genome compositions around CDCA7 genes in Hymenoptera insects are shown. For genome with annotated chromosomes, chromosome numbers (Chr) or linkage group numbers (LG) are indicated at each gene cluster. Gene clusters without chromosome annotation indicate that they are within a same scaffold or contig. Gene locations within each contig are listed in *Figure 7—source data 1*. Dash lines indicate the long linkages not proportionally scaled in the figure. Due to their extraordinarily long sizes, DE-cadherin

*Figure 7 continued on next page*

*Figure 7 continued*

genes (**L**) are not scaled proportionally. Presence and absence of 5mC, CDCA7, HELLS, DNMT1, DNMT3, and UHRF1 in each genome is indicated by filled and open boxes, respectively. Absence of 5mC in *Aphidus gifuensis* (marked with an asterisk) is deduced from the study in *Aphidius ervi* (**Bewick et al., 2017b**), which has an identical presence/absence pattern of the listed genes (**Figure 7—source data 2**).The phylogenetic tree is drawn based on published analysis (**Li et al., 2021**; **Peters et al., 2017**) and TimeTree.

The online version of this article includes the following source data for figure 7:

**Source data 1.** Summary of Hymenoptera genome location for **Figure 7**.

**Source data 2.** Lists of proteins in Hymenoptera species supporting **Figure 7**.

evolutionary preservation of CDCA7 is tightly coupled to the presence of HELLS and DNMT1-UHRF1. The conservation of CDCA7's signature cysteine residues alongside three ICF-associated residues across diverse eukaryote lineages suggests a unique evolutionary conserved role in DNA methylation. Our co-evolution analysis suggests that DNA methylation-related functionalities of CDCA7 and HELLS are inherited from LECA.

The evolutionary coupling of CDCA7, HELLS and DNMT1 is consistent with a proposed role of HELLS in replication-uncoupled DNA methylation maintenance (**Ming et al., 2020**). Commonly, DNA methylation maintenance occurs directly behind the DNA replication fork. Replication-uncoupled DNA methylation maintenance is distinct from this process (**Nishiyama et al., 2020**), and HELLS and CDCA7 may be important for the maintenance of DNA methylation long after the completion of DNA replication, particularly at heterochromatin where chromatin has restricted accessibility (**Ming et al., 2020**). The tighter evolutionary coupling of CDCA7-HELLS to DNMT1 rather than to DNMT3 may also reflect a potential capacity of CDCA7 in sensing DNA methylation, similar to the way that the zf-CXXC domain is sensitive to CpG methylation (**Long et al., 2013**), but in a replication-coupled manner. However, this does not necessarily suggest that the role of CDCA7 is always coupled to maintenance DNA methylation.

The loss of CDCA7 is not always coupled to the loss of DNMT1 or HELLS. In the Hymenoptera clade, CDCA7 loss in the braconid wasps is accompanied with the loss of DNMT1/UHRF1 or the loss of DNMT3. Among these species, it was reported that 5mC DNA methylation is undetectable in *M. demolitor*, which harbors DNMT1, UHRF1 and HELLS but lost DNMT3 and CDCA7 (**Bewick et al., 2017b**). Similarly, in the Coleoptera clade, the red flour beetle *T. castaneum* possesses DNMT1 and HELLS, but lost DNMT3 and CDCA7 (**Figure 6A** and **Figure 1—source data 1**). Since DNMT1 is essential for the embryonic development of *T. castaneum* and CpG DNA methylation is undetectable in this organism (**Schulz et al., 2018**), it has been predicted that DNMT1 has a function independent of DNA methylation in this species. Indeed, species that preserves DNMTs and/or HELLS in the absence of CDCA7 emerge repeatedly during eukaryote evolution, whereas CDCA7 appears to be immediately dispensable in species that have a dampened requirement for DNA methylation. In other words, there is no evolutionary advantage to retain CDCA7 in the absence of DNA methylation, and CDCA7 is almost never maintained in the absence of any DNMTs. Alternatively, could the loss of CDCA7 precede the loss of DNA methylation? If CDCA7 is important to promote DNA methylation maintenance, the loss of CDCA7 may exacerbate an impaired epigenetic environment and stimulate the adaptation of the organism towards DNA methylation-independent epigenetic mechanisms, thereby decreasing the necessity to maintain the DNA methylation system. In this way, the loss of CDCA7 could trigger the subsequent loss of DNMTs (and HELLS), unless these proteins acquired important DNA methylation-independent roles. In line with this scenario, insects, whose DNA methylation is largely limited to gene bodies, possess robust DNA methylation-independent mechanisms (such as piwi-RNA and H3K9me3) to silence transposons (**Bonasio et al., 2012**; **Feng et al., 2010**; **Libbrecht et al., 2016**; **Wang et al., 2013**; **Zemach et al., 2010**). The loss of CDCA7 is more frequently observed in insects than plants and vertebrates, where DNA methylation and HELLS/DDM1 play major roles in silencing transposons. (**Czech et al., 2018**; **Dunican et al., 2013**; **Huang et al., 2004**; **Kato et al., 2003**; **Miura et al., 2001**; **Onishi et al., 2021**; **Osakabe et al., 2021**; **Vongs et al., 1993**). Thus, lowering the demand of DNA methylation at transposable elements might reduce the essentiality of CDCA7 (and then perhaps that of DNMTs), although it is difficult to deduce which evolutionary change occurs first.

Considering the importance of HELLS/DDM1 in silencing transposable elements, it is intriguing that CDCA7, HELLS, and DNMT1 are conserved in many insects, in which transposable elements are generally hypomethylated (**Bonasio et al., 2012**; **Feng et al., 2010**; **Libbrecht et al., 2016**;

*Wang et al., 2013*; *Zemach et al., 2010*). DNMT1 knockout in the clonal raider ant, *Ooceraea biroi*, which has a full set of DNMT1, UHRF1, DNMT3, CDCA7 and HELLS (*Figure 7*), does not cause major developmental defects but leads to failure in reproductive oogenesis and compromised longevity (*Ivasyk et al., 2023*). Similarly, DNMT1 is essential for meiosis (but not for repression of transposable elements) in the milkweed bug *Oncopeltus fasciatus* (*Bewick et al., 2019b*; *Washington et al., 2021*). HELLS and DNMT1 are also important for meiotic progression in mice (*Baumann et al., 2020*; *Spruce et al., 2020*; *Takada et al., 2021*; *Zeng et al., 2011*). In *Neurospora*, DNA methylation is a critical component for trans-sensing homologous chromosomes during meiosis (*Pratt et al., 2004*). Since 5mC is highly mutagenic and DNA methylation patterns are altered during aging (*Lowe et al., 2018*; *Wang et al., 2020*), it is tempting to speculate that the precise maintenance of DNA methylation patterns may act as a hallmark to distinguish between young/healthy DNA and old/mutated (or competitive/pathogenic) DNA during meiosis, perhaps in a way analogous to the role of methylated DNA for mismatch repair in bacteria (*Schofield and Hsieh, 2003*).

The observation that some species retain HELLS but lose CDCA7 (while the reverse is never true) suggests that HELLS can evolve a CDCA7-independent function. Indeed, it has been suggested that the sequence-specific DNA-binding protein PRDM9 recruits HELLS to meiotic chromatin to promote DNA double-strand breaks and recombination (*Imai et al., 2020*; *Spruce et al., 2020*). Unlike CDCA7, clear PRDM9 orthologs are found only in metazoans, and are even lost in some vertebrates such as *Xenopus laevis* and *Gallus gallus* (*Birtle and Ponting, 2006*). Thus, it is plausible that HELLS gained a species-specific CDCA7-independent role during evolution through acquiring a new interaction partner. Another example of this may be found in *S. cerevisiae*, where DNA methylation and CDCA7 are absent and the HELLS homolog Irc5 interacts with cohesin to facilitate its chromatin loading (*Litwin et al., 2017*). In *Neurospora*, where genomic DNA is methylated, the HELLS homolog MUS-30 is not required for DNA methylation but plays a role in DNA damage responses (*Basenko et al., 2016*). We speculate that the role of CDCA7 is evolutionarily more tightly coupled to DNA methylation than HELLS is.

Recently, the role of HELLS in the deposition of the histone variant macroH2A, which compacts chromatin, has been reported in mice (*Ni and Muegge, 2021*; *Ni et al., 2020*; *Xu et al., 2021*). Similarly, in *Arabidopsis*, DDM1 is critical for deposition of H2A.W, which is enriched on heterochromatin, in a manner independent of DNA methylation (*Osakabe et al., 2021*). The role of CDCA7 in the deposition of these H2A variants remains to be tested. HELLS and DDM1 can directly interact with macroH2A and H2A.W, respectively, even in the absence of CDCA7 (*Ni and Muegge, 2021*; *Osakabe et al., 2021*). It is thus possible that HELLS/DDM1 family proteins have an evolutionary conserved function in H2A variant deposition independent of CDCA7 and DNA methylation. However, there is no clear indication of coevolution of HELLS and macroH2A, as macroH2A is largely missing from insects and the chelicerata *Centruroides sculpturatus* has macroH2A (XP_023217082, XP_023212717) but lost HELLS and CDCA7.

Whereas CDCA7-like proteins with class I zf-4CXXC_R1 are evolutionarily coupled to HELLS and DNMT1-UHRF1, other variants of zf-4CXXC_R1 are widespread in eukaryotes except for metazoans, which encode only CDCA7 orthologs and their close paralogs (with the exception of the sponge *A. queenslandica*). Proteins containing a diverged variant of zf-4CXXC_R1 and the JmjC domain, such as IBM1 in *Arabidopsis* and DMM-1 in Neurospora, are broadly found in Viridiplantae and Fungi (*Honda et al., 2010*; *Saze et al., 2008*; *Zhang et al., 2022*). Functional studies of IBM1 and DMM-1 suggested that they contribute to DNA methylation regulation via indirect mechanisms. As IBM1 and DMM-1 do not preserve ICF-associated residues, which are critical for nucleosome binding in CDCA7 (*Jenness et al., 2018*), it is likely that these variants of zf-4CXXC_R1 are adapted to recognize different structural features of the genome and no longer preserve the DNA methylation function of CDCA7 orthologs.

Considering the broad conservation of DNA methylation in vertebrates (*Hemmi et al., 2000*; *Kondilis-Mangum and Wade, 2013*), plants (*Deleris et al., 2016*), prokaryotes (*Beaulaurier et al., 2019*; *Casadesús and Low, 2006*; *Dimitriu et al., 2020*; *Vasu and Nagaraja, 2013*) and Archaea (*Grogan, 2003*; *Hayashi et al., 2021*; *Ishikawa et al., 2005*; *Prangishvili et al., 1985*), along with the existence of SNF2-like proteins (SSO1653) in prokaryotes and Archaea (*Flaus et al., 2006*), we hypothesize that the evolutionary advent of zf-4CXXC_R1-containing CDCA7 was a key step to transmit the

DNA methylation system from the last universal common ancestor (LUCA) to the eukaryotic ancestor with nucleosome-containing genomes.

## Methods

### Key resources table

| Reagent type (species) or resource | Designation | Source or reference | Identifiers | Additional information |
|---|---|---|---|---|
| Software, algorithm | MacVector | MacVector, Inc | Version 16–18 | |
| Software, algorithm | Muscle | https://www.drive5.com/muscle/ | Muscle5.1 | |
| Software, algorithm | IQ-TREE | http://www.iqtree.org/ | Version 2.0.3 and 2.2.2.6 | |
| Software, algorithm | Timetree | http://www.timetree.org/ | Version 5 | |
| Software, algorithm | phyloT | https://phylot.biobyte.de/ | Version 2 | |
| Software, algorithm | iTOL | https://itol.embl.de/ | Version 6 | |
| Software, algorithm | CoPAP | http://copap.tau.ac.il/source.php | | |
| Software, algorithm | ETE Toolkit | http://etetoolkit.org/ | | |
| Software, algorithm | Jalview | https://www.jalview.org/ | Version 2.22.2.7 | |

### Building a curated list of 180 species for analysis of evolutionary co-selection

A list of 180 eukaryote species was manually generated to encompass broad eukaryote evolutionary clades (*Figure 1—source data 1*). Species were included in this list based on two criteria: (i) the identification UBA1 and PCNA homologs, two highly conserved and essential proteins for cell proliferation; and (ii) the identification of more than 6 distinct SNF2 family sequences. Homologs of CDCA7, HELLS, UBA1, and PCNA were identified by BLAST search against the Genbank eukaryote protein database available at National Center for Biotechnology Information using the human protein sequence as a query (NCBI). Homologs of human UHRF1, ZBTB24, SMARCA2/SMARCA4, INO80, RAD54L, EZH2, EED, or Suz12 were also identified based on the RBH criterion. To get a sense of genome assembly level of each genome sequence, we divided 'Total Sequence Length' by 'Contig N50' ('length such that sequence contigs of this length or longer include half the bases of the assembly'; https://www.ncbi.nlm.nih.gov/assembly/help/). In the species whose genome assembly level is labeled as 'complete', this value is close to the total number of chromosomes or linkage groups. As such, as a rule of thumb, we arbitrarily defined the genome assembly 'preliminary', if this value is larger than 100. In *Figure 5*, these species with preliminary-level genome assembly were noted as boxes with dotted outlines.

CDCA7 homolog identification and annotation BLAST search was conducted using human CDCA7 (NP_114148) as the search query against NCBI protein database. The obtained list of CDCA7 homologs was classified based on the conservation of eleven cysteine and three ICF-associated residues in the zf-4CXXC_R1 domain, as described in Results. This classification was further validated based on their clustering in a phylogenetic tree built from the CLUSTALW alignment of the zf-4CXXC_R1 domain identified by NCBI conserved domains search (*Higgins and Sharp, 1988*; *Lu et al., 2020*; *Thompson et al., 1994*; *Figure 2—figure supplement 1*), using MacVector (MacVector, Inc). Jalview was used to color-code amino acids based on conservation and amino acid types (*Waterhouse et al., 2009*). The cluster of class I zf-4CXXC_R1 domain-containing proteins (where all three ICF-associated residues are conserved) was segregated from other variants of zf-4CXXC_R1-containing proteins except for the moss *Physcomitrium* XP_024393821 and green algae *Bathycoccus* XP_007512509 and *Chlamydomonas* GAX81623, which has class II zf-4CXXC-R1 (where the ICF-associated glycine residue is substituted). To further assess this validation, another phylogenetic tree was built by the optimal maximum likelihood-based model selected by IQ-TREE 2.2.2.6 (*Minh et al., 2020*), using the zf-4CXXC_R1 domain alignment generated and selected by Muscle v5 (*Figure 2—source data 1*; *Edgar, 2022*). A consensus tree was then constructed from 1000 bootstrap trees using UFBoot2 (*Figure 2—source data 2*; *Hoang et al., 2018*). The phylogenetic tree built by this second method deviated all class II and other zf-4CXXC_R1 variants from CDCA7 orthologs with class I zf-4CXXC_R1 (*Figure 2—source data 2*).

## HELLS homolog identification and annotation

HELLS homologs were first identified according to the RBH criterion. Briefly, a BLAST search was conducted using human HELLS as the query sequence, after which protein sequences of obtained top hits (or secondary hits, if necessary) in each search were used as a query sequence to conduct reciprocal BLAST search against the *Homo sapiens* protein database. If the top hit in the reciprocal search returned the original input sequence (i.e. human HELLS), it was temporarily annotated as an orthologous protein. If HELLS showed up as a next best hit, it is temporarily listed as a "potential HELLS ortholog". To further validate the identified HELLS orthologs, full length amino acid sequences of these proteins were aligned using CLUSTALW in MacVector with homologs of the Snf2 family proteins SMARCA2/SMARCA4, CHD1/CHD3/CHD7, ISWI, RAD54L, ATRX, HLTF, TTF2, SHPRH, INO80, SMARCAD1, SWR1, MOT1, ERCC6, and SMARCAL1, which were also identified and temporarily annotated with a similar reciprocal BLAST search methods. The phylogenetic tree generated by this full-length alignment separated the clade containing HELLS, DDM1, INO80, SMARCA2/SMARCA4, CHD1/CDH3/CHD7, and ISWI from other Snf2 family proteins (*Figure 5—figure supplement 2*). This alignment was used to define the conserved SNF2 domain and variable linker regions in the putative HELLS orthologs and other homologous SNF2-family proteins (INO80, SMARCA2/SMARCA4, CHD1/CDH3/CHD7, and ISWI). The variable linker regions were then removed from selected proteins for each kingdom/phylum to conduct the secondary CLUSTALW alignment in MacVector, from which a phylogenetic tree was generated (*Figure 5—figure supplement 3*). A distinct cluster of HELLS and other SNF2 family proteins can be identified from the phylogenetic tree, hence confirming that the annotation of HELLS orthologs based on the reciprocal BLAST search method is reasonable. Exceptions are *Leucosporidium creatinivorum* ORY88017 and ORY88018, which did not cluster within the HELLS clade of the phylogenetic tree. However, we decided to annotate *L. creatinivorum* ORY88017 and ORY88018 as HELLS orthologs since among other *L. creatinivorum* SNF2-family proteins in this species these two proteins are most similar to human HELLS while clear orthologs of other *L. creatinivorum* SNF2 family proteins, CHD1 (ORY55731), ISWI (ORY89162), SMARCA2/4 (ORY76015), SRCAP/SWR1 (ORY90750), and INO80 (ORY91599), can be identified. To further validate the phylogenetic tree generated in *Figure 5—figure supplement 3*, another phylogenetic tree was built by the optimal maximum likelihood-based model selected by IQ-TREE 2.2.2.6 (*Minh et al., 2020*), using the alignment generated and selected by Muscle v5 (*Figure 5—source data 1*; *Edgar, 2022*). A consensus tree was then constructed from 1000 bootstrap trees using UFBoot2 (*Figure 5—source data 2*; *Hoang et al., 2018*). The topology of the phylogenetic tree build by this second method was consistent with the original tree shown in *Figure 5—figure supplement 3*.

## DNMT homolog identification and annotation

Proteins with a DNA methyltransferase domain were identified with BLAST searches using human DNMT1 and DNMT3A. Additional BLAST searches were conducted using human DNMT2, *C. neoformans* DNMT5, *N. crassa* Dim-2 and DNMT4, *Thalassiosira pseudonana* DNMT6 (XP_002287631; *Huff and Zilberman, 2014*), and *Kwoniella mangroviensis* DNMTX (XP_019001759; *Catania et al., 2020*). DNMT domains were extracted from selected proteins that represent each kingdom/phylum and DNMT type (including ambiguous classes) based on a NCBI conserved domains search and were aligned with CLUSTALW to build a phylogenetic tree in MacVector. To further assess this validation, another phylogenetic tree was built by the optimal maximum likelihood-based model selected by IQ-TREE 2.0.3 (*Minh et al., 2020*), using the alignment generated and selected by Muscle v5 (; *Edgar, 2022*). A consensus tree was then constructed from 1000 bootstrap trees using UFBoot2 (; *Hoang et al., 2018*). The topology of the phylogenetic tree built by this second method was largely consistent with the original tree except for placement of orphan DNMTs. In both trees, *Vitrella brassicaformis* CEM15752, *Fragilariopsis cylindrus* OEU08290, OEU15938, *Thalassiosira pseudonana* XP_002295472, and *Tribonema minus* KAG5185060 form a clade, which, only in the IQ-TREE generated tree, splits from the canonical DNMT4 clade. As proteins in this clade have been previously annotated as DNMT4 (*Huff and Zilberman, 2014*), we followed this classification and present this IQ-TREE-based phylogenetic tree of DNMTs in *Figure 5—figure supplement 4*, which was illustrated using the ETE3 toolkit.

## CoPAP

The published method was used (*Cohen et al., 2013*). The curated list of orthologous proteins listed in *Figure 1—source data 1* was first used to generate a presence-absence FASTA file. Next, a phylogenetic species tree was generated from all orthologous protein sequences listed in *Figure 1—source data 1* using the ETE3 toolkit. For this, protein sequences were retrieved using the rentrez Bioconductor package and exported to a FASTA file alongside a COG file containing gene to orthologous group mappings. ETE3 was used with the parameters *-w clustalo_default-trimal01-none-none* and *-m cog_all-alg_concat_default-fasttree_default* and the resulting tree exported in Newark format. CoPAP was run using default parameters and results visualized using Cytoscape. Code and files required for CoPAP input generation as well CoPAP parameters and output results can be found in our Github repository (https://github.com/RockefellerUniversity/Copap_Analysis copy archived at *Carroll, 2023*). A method to calculate *p*-value for CoPAP was described previously (*Cohen et al., 2012*). Briefly, for each pair of tested genes, Pearson's correlation coefficient was computed. Parametric bootstrapping was used to compute a *p*-value by comparing it with a simulated correlation coefficient calculated based on a null distribution of independently evolving pairs with a comparable exchangeability (a value reporting the likelihood of gene gain and loss events across the tree).

As negative and positive controls for the CoPAP analysis, we identified several well-conserved protein orthologs across the panel of 180 eukaryotic species, including Snf2-like proteins SMARCA2/SMARCA4, INO80, and RAD54L (*Flaus et al., 2006*), as well as subunits of the polycomb repressive complex 2 (PRC2), which plays an evolutionary conserved role in gene repression via deposition of the H3K27me3 mark. PRC2 is conserved in species where DNMTs are absent (including in *D. melanogaster* and *C. elegans*) but is frequently lost particularly in several lineages of SAR and Fungi (*Sharaf et al., 2022*). Among the four core subunits of PRC2, we focused on the catalytic subunit EZH1/2, EED, and SUZ12, since the fourth subunit RbAp46/48 has a PRC2-independent role (*Margueron and Reinberg, 2011*). We are aware that the reciprocal BLAST search missed previously reported highly divergent functional orthologs of SUZ12 in *Neurospora* (*Jamieson et al., 2013*), and EED and Suz12 in *Paramecium* (*Miró-Pina et al., 2022*). However, we did not attempt to use these divergent homologs of EED and SUZ12 as baits to expand our search in order to consistently apply our homology-based definition of orthologs.

## Hymenoptera synteny analysis

The mapping of gene loci is based on the information available on the Genome Data Viewer (https://www.ncbi.nlm.nih.gov/genome/gdv). Genome positions of listed genes are summarized in *Figure 7—source data 1*.

## Artworks

Artworks of species images were obtained from https://www.phylopic.org/, of which images of Daphnia, Platyhelminthes, Tribolium and Volvox were generated by Mathilde Cordellier, Christopher Laumer/T. Michael Keesey, Gregor Bucher/Max Farnworth and Matt-Crook, respectively.

# Acknowledgements

We thank Daniel Kronauer and Rochelle Shih for critical reading of the manuscript, and D Kronauer, Li Zhao, Junhui Peng and Erick Jarvis for helpful discussion. We thank Yasuhiro Arimura and Hideaki Konishi for their assistance in multiple sequence alignment and phylogenetic tree generation. The research by Q J was in part executed through Chemers Neustein Summer Undergraduate Research Fellowship (SURF) Program at the Rockefeller University. Funding This work was supported by National Institutes of Health Grants R35GM132111 to HF, and the Women & Science Postdoctoral Fellowship Program at The Rockefeller University to IEW.

## Additional information

### Funding

| Funder | Grant reference number | Author |
|---|---|---|
| National Institutes of Health | R35GM132111 | Hironori Funabiki |
| The Rockefeller University | Women & Science Postdoctoral Fellowship | Isabel E Wassing |

The funders had no role in study design, data collection and interpretation, or the decision to submit the work for publication.

### Author contributions

Hironori Funabiki, Conceptualization, Data curation, Supervision, Funding acquisition, Investigation, Visualization, Writing – original draft, Project administration, Writing – review and editing; Isabel E Wassing, Supervision, Funding acquisition, Investigation, Writing – review and editing; Qingyuan Jia, Data curation, Project administration, Investigation, Visualization, Writing - review and editing; Ji-Dung Luo, Thomas Carroll, Data curation, Project administration, Writing - review and editing

### Author ORCIDs

Hironori Funabiki ⓘ https://orcid.org/0000-0003-4831-4087
Ji-Dung Luo ⓘ http://orcid.org/0000-0003-0150-1440

Reviewer #1 (Public Review): https://doi.org/10.7554/eLife.86721.4.sa1
Reviewer #2 (Public Review): https://doi.org/10.7554/eLife.86721.4.sa2
Author Response https://doi.org/10.7554/eLife.86721.4.sa3

## Additional files

### Supplementary files
• MDAR checklist

### Data availability

Code and files required for CoPAP input generation as well CoPAP parameters and output results can be found in our Github repository (https://github.com/RockefellerUniversity/Copap_Analysis copy archived at *Carroll, 2023*). All supporting data are included as Source Data files for Figure 1, Figure 2, Figure 5 and Figure 7.

The following previously published datasets were used:

| Author(s) | Year | Dataset title | Dataset URL | Database and Identifier |
|---|---|---|---|---|
| Murali S, Richards S, Bandaranaike D, Bellair M, Blankenburg K, Chao H, Dinh H, Doddapaneni H, Dugan-Rocha S, Elkadiri S, Gnanaolivu R, Hernandez B, Skinner E, Javaid M, Lee S, Li M, Ming W, Munidasa M, Muniz J, Nguyen L, Hughes D, Osuji N, L-L Pu, Puazo M, Qu C, Quiroz J, Raj R, Weissenberger G, Xin Y, Zou X, Han Y, Worley K, Muzny GR | 2017 | Genome assembly Cflo_2.0 | https://www.ncbi.nlm.nih.gov/datasets/genome/GCF_000648655.2/ | NCBI RefSeq assembly, GCF_000648655.2 |
| Dalla Benetta E, Antoshechkin I, Yang T, Nguyen HQM, Ferree PM, Akbari OS | 2020 | Genome assembly Nvit_psr_1.1 | https://www.ncbi.nlm.nih.gov/datasets/genome/GCF_009193385.2/ | NCBI RefSeq assembly, GCF_009193385.2 |
| Burke GR, Walden KK, Whitfield JB, Robertson HM, Strand MR | 2014 | Genome assembly Mdem2 | https://www.ncbi.nlm.nih.gov/datasets/genome/GCF_000572035.2/ | NCBI RefSeq assembly, GCF_000572035.2 |
| Geib SM, Liang GH, Murphy TD, Sim SB | 2017 | Genome assembly ASM80636v1 | https://www.ncbi.nlm.nih.gov/datasets/genome/GCF_000806365.1/ | NCBI RefSeq assembly, GCF_000806365.1 |
| Patalano S, Vlasova A, Wyatt C, Ewels P, Camara F, Ferreira PG, Asher CL, Jurkowski TP, Segonds-Pichon A, Bachman M, González-Navarrete I, Minoche AE, Krueger F, Lowy E, Marcet-Houben M, Rodriguez-Ales JL, Nascimento FS, Balasubramanian S, Gabaldon T, Tarver JE, Andrews S, Himmelbauer H, Hughes WO, Guigó R, Reik W, Sumner S | 2015 | Genome assembly ASM131383v1 | https://www.ncbi.nlm.nih.gov/datasets/genome/GCF_001313835.1/ | NCBI RefSeq assembly, GCF_001313835.1 |
| McKenzie SK, Kronauer DJC | 2018 | Genome assembly Obir_v5.4 | https://www.ncbi.nlm.nih.gov/datasets/genome/GCF_003672135.1/ | NCBI RefSeq assembly, GCF_003672135.1 |
| Pan H, Kapheim K | 2015 | Genome assembly ASM126327v1 | https://www.ncbi.nlm.nih.gov/datasets/genome/GCF_001263275.1/ | NCBI RefSeq assembly, GCF_001263275.1 |
| Wellcome Sanger Tree of Life Programme | 2022 | Genome assembly iyBomTerr1.2 | https://www.ncbi.nlm.nih.gov/datasets/genome/GCF_910591885.1/ | NCBI RefSeq assembly, GCF_910591885.1 |

*Continued on next page*

*Continued*

| Author(s) | Year | Dataset title | Dataset URL | Database and Identifier |
|---|---|---|---|---|
| Murali S, Richards S, Bandaranaike D, Bellair M, Blankenburg K, Chao H, Dinh H, Doddapaneni H, Dugan-Rocha S, Elkadiri S, Gnanaolivu R, Hernandez B, Skinner E, Javaid M, Lee S, Li M, Ming W, Munidasa M, Muniz J, Nguyen L, Hughes D, Osuji N, L-L Pu, Puazo M, Qu C, Quiroz J, Raj R, Weissenberger G, Xin Y, Zou X, Han Y, Worley K, Muzny D, Gibbs R | 2017 | Genome assembly Oabi_2.0 | https://www.ncbi.nlm.nih.gov/datasets/genome/GCF_000612105.2/ | NCBI RefSeq assembly, GCF_000612105.2 |
| Burke GR, Simmonds TJ, Geib SM | 2022 | Genome assembly ASM1945775v1 | https://www.ncbi.nlm.nih.gov/datasets/genome/GCF_019457755.1/ | NCBI RefSeq assembly, GCF_019457755.1 |
| Du Z | 2020 | Genome assembly ASM1490517v1 | https://www.ncbi.nlm.nih.gov/datasets/genome/GCF_014905175.1/ | NCBI RefSeq assembly, GCF_014905175.1 |
| Pinto BJ, Weis JJ, Gamble T, Ode PJ, Paul R, Zaspel JM | 2021 | Genome assembly MPM_Cglom_v2.3 | https://www.ncbi.nlm.nih.gov/datasets/genome/GCF_020080835.1/ | NCBI RefSeq assembly, GCF_020080835.1 |

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
