## [Editor Report · eLife assessment]

This **important** manuscript reveals signatures of co-evolution of two nucleosome remodeling factors, Lsh/HELLS and CDCA7, which are involved in the regulation of eukaryotic DNA methylation. The results suggest that the roles for the two factors in DNA methylation maintenance pathways can be traced back to the last eukaryotic common ancestor and that the CDC7A-HELLS-DNMT axis shaped the evolutionary retention of DNA methylation in eukaryotes. The **solid** evolutionary analyses form a strong basis for experimental follow-up studies. The work should be of interest to colleagues in the fields of evolutionary biology, chromatin biology and genome biology.

---

## [Referee Report · Reviewer #1 (Public Review)]

Funabiki et al, performed a co-evolutionary analysis of Lsh/HELLS and CDCA7, two factors with links to DNA methylation pathways in mammals, amphibia and fish. The authors suggest that conserved roles for the two factors in DNA methylation maintenance pathways can be traced back to the last eukaryotic common ancestor. Overall, the findings are important and the results could be useful for researchers studying DNA methylation pathways in many different organisms.

---

## [Referee Report · Reviewer #2 (Public Review)]

In this manuscript, Funabiki and colleagues investigated the co-evolution of DNA methylation and nucleosome remolding in eukaryotes. This study is motivated by several observations: (1) despite being ancestrally derived, many eukaryotes lost DNA methylation and/or DNA methyltransferases; (2) over many genomic loci, the establishment and maintenance of DNA methylation relies on a conserved nucleosome remodeling complex composed of CDCA7 and HELLS; (3) it remains unknown if/how this functional link influenced the evolution of DNA methylation. The authors hypothesize that if CDCA7-HELLS function was required for DNA methylation in the last eukaryote common ancestor, this should be accompanied by signatures of co-evolution during eukaryote radiation.

To test this hypothesis, they first set out to investigate the presence/absence of putative functional orthologs of CDCA7, HELLS and DNMTs across major eukaryotic clades. They succeed in identifying homologs of these genes in all clades spanning 180 species. To annotate putative functional orthologs, they use similarity over key functional domains and residues - such as ICF related mutations for CDCA7 and SNF2 domains for HELLS - as well as maximum likelihood phylogenetic analyses. Using established eukaryote phylogenies, the authors conclude that the CDCA7-HELLS-DNMT axis arose in the last common ancestor to all eukaryotes. Importantly, they found recurrent loss events of CDCA7-HELLS-DNMT in at least 40 eukaryotic species, most of them lacking DNA methylation.

Having identified these factors, they successfully identify signatures of co-evolution between DNMTs, CDCA7 and HELLS using CoPAP analysis - a probabilistic model inferring the likelihood of interactions between genes given a set of presence/absence patterns. As a control, such interactions are not detected with other remodelers or chromatin modifying pathways also found across eukaryotes. Expanding on this analysis, the authors found that CDCA7 was more likely to be lost in species without DNA methylation.

In conclusion, the authors suggest that the CDCA7-HELLS-DNMT axis is ancestral in eukaryotes and raise the hypothesis that CDCA7 becomes quickly dispensable upon the loss of DNA methylation and/or that CDCA7 might be the first step toward the switch from DNA methylation-based genome regulation to other modes.

The data and analyses reported are significant and solid. Overall, this work is a conceptual advance in our understanding of the evolutionary coupling between nucleosome remolding and DNA methylation. It also provides a useful resource to study the early origins of DNA methylation related molecular process. Finally, it brings forward the interesting hypothesis that since eukaryotes are faced with the challenge of performing DNA methylation in the context of nucleosome packed DNA, loosing factors such as CDCA7-HELLS likely led to recurrent innovations in chromatin-based genome regulation.

Strengths:

- The hypothesis linking nucleosome remodeling and the evolution of DNA methylation.

- Deep mapping of DNA methylation related process in eukaryotes.

- Identification and evolutionary trajectories of novel homologs/orthologs of CDCA7.

- Identification of CDCA7-HELLS-DNMT co-evolution across eukaryotes.

---

## [Author Response]

The following is the authors’ response to the previous reviews

**Reviewer #2 (Recommendations For The Authors):**
1. Overall, the novel phylogenetic analyses presented are satisfactory. With this new piece of information in hand, I would suggest using maximum-likelihood analyses as the major evidence supporting ortholog annotations. In fact, it would be best advised to add the bootstrap support analyses (perhaps over new trees) to the phylogenies presented in the supplement.

Thank you for suggestion. Although it would make sense to present phylogenetic trees constructed by maximum-likelihood analyses, we decided to keep the original trees (forCDCA7 and HELLS) in supplemental figures for an aesthetic reason. For example, forCDCA7/zf-4CXXC_R tree made by maximum likelihood method *Hif2_data2_zf4CXXC_R1_iqtree.txt, it would have been easier to visualize if the plant CDCA7 clade was positioned at the bottom, not the top, of the tree, as the topology was identical in both cases. Unfortunately, as the calculated result randomly put plant CDCA7 clade at the top, plant CDCA7 clade appears to be separated from the clades representing the rest of CDCA7 homologs. While we could manually adjust this in the final drawing, we wanted to avoid that.

1. There are still a few places in the main text where RBH - and is associated E-value - is used as evidence of orthology. As mentioned in my original review, this is evidence for homology, not orthology. Please make sure to amend the final text (for example in the first paragraph of the result section).

We concurred and amended the manuscript following this recommendation.

1. We agree with reviewer 1 that part of the functional considerations outside of the human and frog example should be softened, or clearly labelled as an hypothesis - which is now supported by this interesting study

I assume that this is related to Introduction of CDCA7. As this study defined CDCA7 homologs in result section We believe that this point has been addressed in our last revision.

1. In addition, make sure to indicate in the main text state the point about DNMT3 nomenclature (w.r.t. DRM).

In page 10, we added a sentence below to clarify this point.

“In this report, we call a protein DNMT3 if it clusters into the clade including metazoan DNMT3, plant DNMT3, and DRM.”